# Oscillations of Delta-like1 regulate the balance between differentiation and maintenance of muscle stem cells

Yao Zhang [1,9] ✉, Ines Lahmann[1,9], Katharina Baum [2,7], Hiromi Shimojo[3,8], Philippos Mourikis[4], Jana Wolf [2,5], Ryoichiro Kageyama[3] & Carmen Birchmeier [1,6] ✉

Cell-cell interactions mediated by Notch are critical for the maintenance of skeletal muscle stem cells. However, dynamics, cellular source and identity of functional Notch ligands during expansion of the stem cell pool in muscle growth and regeneration remain poorly characterized. Here we demonstrate that oscillating Delta-like 1 (Dll1) produced by myogenic cells is an indispensable Notch ligand for self-renewal of muscle stem cells in mice. Dll1 expression is controlled by the Notch target Hes1 and the muscle regulatory factor MyoD. Consistent with our mathematical model, our experimental analyses show that Hes1 acts as the oscillatory pacemaker, whereas MyoD regulates robust *Dll1* expression. Interfering with Dll1 oscillations without changing its overall expression level impairs self-renewal, resulting in premature differentiation of muscle stem cells during muscle growth and regeneration. We conclude that the oscillatory Dll1 input into Notch signaling ensures the equilibrium between self-renewal and differentiation in myogenic cell communities.

[1] Developmental Biology/Signal Transduction, Max-Delbrück-Center for Molecular Medicine, Berlin, Germany. [2] Mathematical Modelling of Cellular Processes, Max-Delbrück-Center for Molecular Medicine, Berlin, Germany. [3] Institute for Frontier Life and Medical Sciences, Kyoto University, Kyoto, Japan. [4] Univ Paris Est Creteil, INSERM, IMRB, Creteil, France. [5] Department of Mathematics and Computer Science, Free University Berlin, Berlin, Germany. [6] Neurowissenschaftliches Forschungzentrum, NeuroCure Cluster of Excellence, Charité-Universitätsmedizin Berlin, Berlin, Germany. [7] Present address: Hasso Plattner Institute, Digital Engineering Faculty, University of Potsdam, Potsdam, Germany. [8] Present address: Graduate School of Frontier Biosciences, Osaka University, Osaka, Japan. [9] These authors contributed equally: Yao Zhang, Ines Lahmann. ✉email: Yao.Zhang@mdc-berlin.de; cbirch@mdc-berlin.de

Cell-cell interactions mediated by Notch signaling control development and tissue maintenance. Notch ligands like Delta-like1 (Dll1), expressed on signal-sending cells, activate the Notch receptors on neighboring signal-receiving cells, where transcriptional repressors of the Hes/Hey family are induced[1–4]. Notch signaling has many known functions, among them regulating self-renewal and differentiation of skeletal muscle progenitor and stem cells. Genetic ablation of Notch signaling in mice, either by mutating the gene encoding Dll1, or the transcriptional mediator of Notch signals, RBPj, results in upregulation of the myogenic basic helix loop helix (bHLH) factor MyoD, premature myogenic differentiation, and depletion of the muscle progenitor/stem cell pool[5–9]. This drastic differentiation phenotype is rescued by mutating *MyoD*, indicating that a major role of Notch is to repress *MyoD*[6]. Accordingly, forced Notch activation suppresses *MyoD* and myogenic differentiation[7,10–14]. Muscle progenitor cells proliferate during development and in the postnatal period, whereas in the adult most muscle stem cells are quiescent and are only re-activated to proliferate when the muscle is injured[15]. Muscle progenitors and quiescent stem cells express Pax7 and/or Pax3, and can either self-renew or give rise to differentiating myoblasts needed for muscle growth and repair[16,17]. Muscle stem cells express myogenic genes like *MyoD* when they are activated, and MyoD has a dual role in controlling proliferation of activated stem cells and the initiation of the muscle-specific differentiation program[18–20]. MyoG drives terminal differentiation and is induced when the cells exit the cell cycle[18,19,21]. Several molecular mechanisms such as the direct repression of *MyoD* and *MyoG* by Hes/Hey factors have been implicated in Notch-dependent control of myogenesis[6,22–27].

Dynamic expression of regulatory factors can encode distinct information and result in different biological outcomes. For instance, oscillatory or sustained Ascl1 expression determines whether a cell will remain a neural progenitor or differentiate, and oscillatory or sustained signaling of p53 controls distinct pathways that affect cell fate[28–30]. Moreover, oscillatory signals allow for more stable network responses than impulse signals that are more difficult to distinguish from noise[31]. The expression of Notch signaling components and their downstream targets oscillates in several cell types, for instance in myogenic and neuronal stem cells and in cells of the presomitic mesoderm[25,30,32]. Oscillatory periods are species dependent, with oscillatory periods of 2–3 and 5–6 h in murine and human cells, respectively[33]. In the myogenic lineage, Hes1 oscillations drive MyoD oscillations[25]. Interestingly, MyoD expression dynamics are distinct in self-renewing or differentiating myogenic cells. Stable oscillatory MyoD expression is observed during amplification of the activated muscle stem cell pool, whereas unstable oscillations and sustained MyoD expression occur during terminal differentiation, suggesting that oscillatory versus sustained expression of MyoD determines myogenic fate[25].

Dll1 expression is known to oscillate in the presomitic mesoderm, neuronal and pancreatic progenitor cells[34–36]. In situ hybridization experiments provide only a snapshot of expression dynamics, but demonstrated that *Dll1* is expressed in a salt and pepper pattern in the developing muscle (http://www.eurexpress.org). This raises the possibility that Dll1 is also dynamically produced in myogenic cells. Mathematical modeling and experimental evidence revealed several prerequisites for stable oscillations of Notch signaling components: (i) negative feedback regulating transcription; (ii) short half-lives of oscillating mRNAs and the proteins they encode; (iii) specific delay times between transcription and protein production[36–40]. The delay depends on variables like the time required for transcription, which can be experimentally manipulated by changing transcript length. For instance, inserting a cDNA encoding a Dll1-luciferase fusion protein into the *Dll1* locus (*Dll1type2* mutant) increases the length of the *Dll1* transcription unit, thus prolonging the delay time between *Dll1* transcription and translation. This results in unstable Dll1 oscillations and severely affects somitogenesis and the timing of neuronal differentiation[36].

In this study, we investigate Dll1 expression dynamics in muscle progenitor and stem cells. We observe that Dll1 is expressed in an oscillatory manner in muscle progenitor cells and in activated, but not quiescent stem cells of the adult muscle. We show that MyoD and Hes1 directly control Dll1 expression by enhancing and repressing *Dll1* transcription, respectively. Downregulation of Hes1 precludes stable Dll1 oscillations and increases Dll1 levels, whereas ablation of MyoD reduces Dll1 expression levels without interfering with oscillatory expression. To study the functional consequences of Dll1 oscillations in myogenesis, we use the *Dll1type2* allele that fails to oscillate in presomitic mesoderm and neuronal progenitors due to an increased delay time between transcription initiation and translation. Myogenic cells that carry the homozygous *Dll1type2* allele express normal levels of Dll1, but the expression dynamics is altered and stable oscillations are no longer observed in communities of cells contacting each other. This results in a higher propensity for muscle stem cells to undergo terminal differentiation and impaired self-renewal of the developing and adult stem cell pool. The skewed balance between self-renewal and differentiation severely affects muscle growth and repair. Our data demonstrate that not only the level but also the dynamics of Dll1 expression encodes information. Thus, oscillatory Dll1 expression in communities of myogenic cells is required for the appropriate coordination of self-renewal and terminal myogenic differentiation.

## Results

**Dll1 produced by muscle progenitor and activated muscle stem cells controls the self-renewal of neighboring cells.** Published RNAseq and microarray data indicate that *Dll1* is expressed in adult muscle stem cells[41–43]. To precisely distinguish between quiescent, activated, and differentiating stem cells, we used mice carrying a *Dll1-luciferase* fusion gene[36] (*Dll1luc*) to define which cells of the myogenic lineage produce Dll1 protein (see Supplementary Fig. 1a, b for a scheme of *Dll1luc* and for data that show that the *Dll1luc* allele does not interfere with myogenesis). We isolated myofibers together with the associated muscle stem cells and used anti-luciferase antibodies to assess Dll1 expression. When fibers are freshly isolated, associated stem cells are quiescent. Upon culture as floating fibers, stem cells remain associated with the fibers. They also display a stereotypic behavior, become activated, and go through a first cell division after around 40 h of culture. Subsequent divisions are faster and the colony size becomes heterogenous. We could not detect Dll1-luciferase protein in freshly isolated fibers or the associated quiescent Pax7+/MyoD- muscle stem cells (Fig. 1a; quantified in Supplementary Fig. 1c). After 24 h of culture, single activated muscle stem cells co-expressing MyoD and Pax7 were observed, and Dll1-luciferase protein was detected in such activated stem cells (Fig. 1a; quantified in Supplementary Fig. 1c). After 72 h in culture, fiber-associated stem cells formed small colonies containing cells that express Pax7, cells co-expressing MyoD and Pax7, or differentiating cells expressing MyoG. MyoD+ and MyoG+ cells expressed Dll1-luciferase, and levels in MyoG+ cells were higher than the ones in MyoD+ cells (Fig. 1a where an unusually small colony is shown to simplify the display; MyoG+ and MyoD+ cells expressing Dll1-luciferase are quantified in Supplementary Fig. 1c). Next, we verified *Dll1* expression in vivo using single-molecule fluorescence in situ hybridization (smFISH). In the

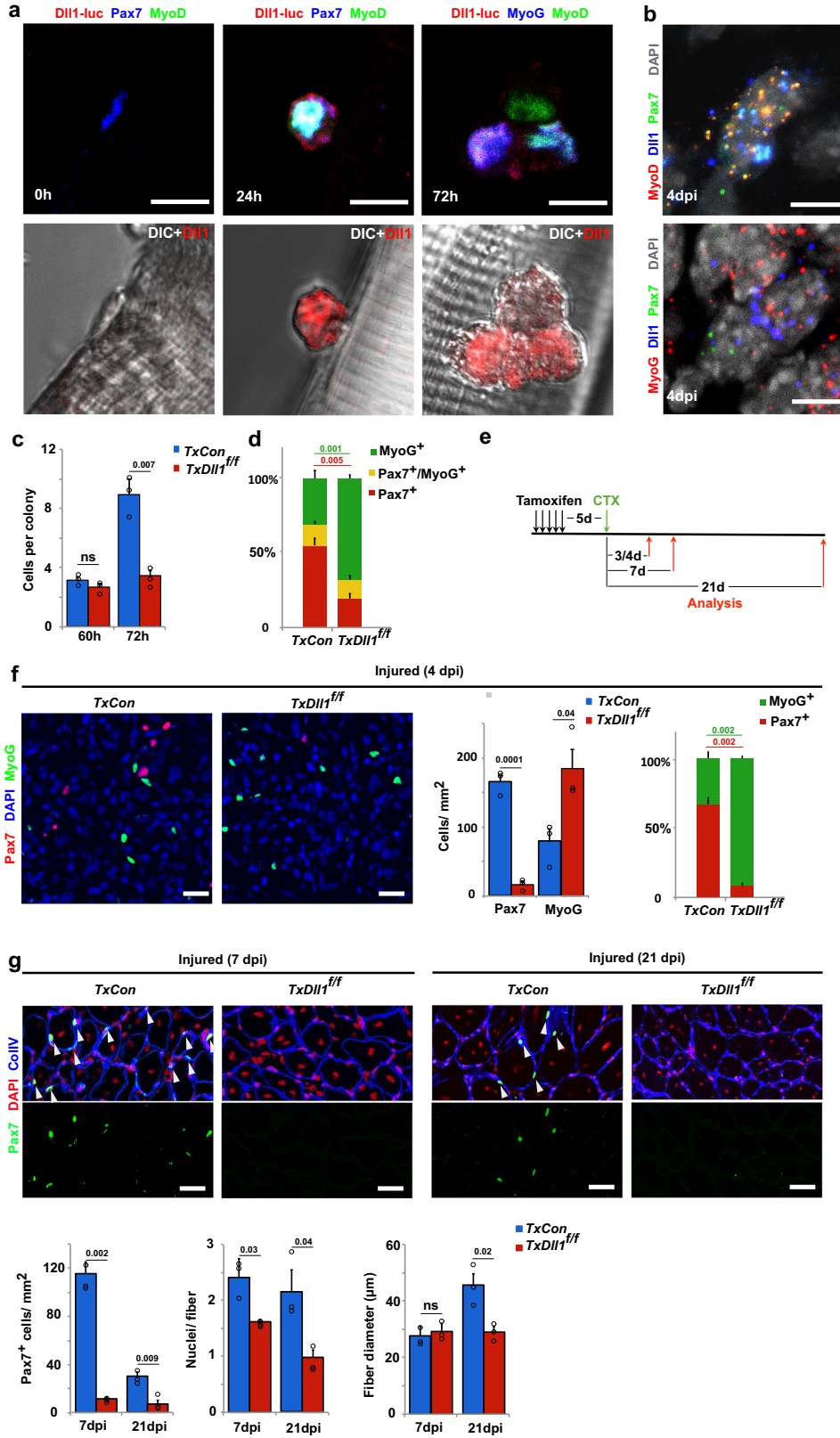

uninjured muscle, quiescent Pax7+/MyoD− stem cells were Dll1 negative, and *Dll1* transcripts were not detected in muscle fibers (Supplementary Fig. 1d, f). After muscle injury, *Dll1* transcripts were present in activated MyoD+/Pax7+ stem cells and in differentiating MyoG+ cells (Fig. 1b and Supplementary Fig. 1d–f). Moreover, in myogenic cells during development, *Dll1* is

expressed in a salt and pepper pattern and present in MyoD+ and MyoG+ cells (Supplementary Fig. 2a).

We asked whether Dll1 produced by activated satellite cells is of functional importance. We used a genetic strategy to introduce a *Dll1* null mutation in adult muscle stem cells, i.e. *Pax7^{IRE-SCreERT2};Dll1^{flox/flox}* mice treated with tamoxifen (hereafter called

**Fig. 1 Dll1 expressed by activated and differentiating muscle stem cells controls the maintenance of the stem cell pool. a** Upper panels: Immunofluorescence analysis of muscle stem cells associated with fibers from mice carrying the *Dll1^luc* allele; myofibers were freshly isolated (0 h), or cultured for 24 and 72 h as indicated. Lower Panels: corresponding differential interference contrast (DIC) images. Dll1-Luc (anti-luciferase; red), Pax7 or MyoG (blue), MyoD (green); $n = 4$ animals. **b** RNAscope analysis of a regenerating tibialis anterior (TA) muscle at 4 days post injury (dpi); MyoD or MyoG (red), Dll1 (blue), Pax7 (green); DAPI was used as counter stain (white); $n = 5$ animals. **c** Quantification of colony size of muscle stem cells on fibers from control (*TxCon*; blue bars) and Dll1 mutant (*TxDll1^f/f*; red bars) animals; fibers were cultured for 60 and 72 h. $n = 3$ animals. **d** Quantification of cells that express Pax7 (red bars), Pax7 and MyoG (yellow bars), MyoG (green bars) in colonies associated with myofibers from control (*TxCon*) and Dll1 null mutant (*TxDll1^f/f*) animals cultured for 72 h; $n = 3$ animals. **e** Schematic outline of the regeneration experiment shown in f,g. The Dll1 null mutation was induced by tamoxifen (*TxDll1^f/f* mutant mice); black arrows: tamoxifen injections; green arrow: cardiotoxin (CTX) treatment; red arrows: analysis of muscle regeneration. **f** Immunohistological analysis of cells expressing Pax7 (red) and MyoG (green) in the injured muscle of control (*TxCon*) and Dll1 null mutant (*TxDll1^f/f*) animals at 4dpi (left panels). Quantifications of Pax7+ and MyoG+ cells (right panels). Quantified were the numbers of Pax7+ or MyoG+ cells/mm$^2$ in *TxCon* (blue bars) and *TxDll1^f/f* (red bars) animals, which is also displayed as the proportion of Pax7+ (red) and MyoG+ (green) cells ($n = 3$ animals). **g** Immunohistological analysis of cells expressing Pax7 (green), Collagen IV (ColIV, blue) in the injured muscle of control (*TxCon*) and Dll1 null mutant (*TxDll1^f/f*) animals at 7dpi (left) and 21 dpi (right); DAPI (red) is used as counterstain. Quantifications of Pax7+ cells/mm$^2$, nuclei/fiber and fiber diameter are shown below (*TxCon*, blue bars and *TxDll1^f/f*, red bars); $n = 3$ animals. Scale bars, 10 μm (**a, b**) and 50 μm (f,g). Data are presented as mean values ± SEM. Exact *p* values are indicated, unpaired two-sided *t*-test.

*TxDll1^f/f* animals; *Pax7^IRESCreERT2;Dll1^luc/flox* treated with tamoxifen were used as controls and are hereafter called *TxCon*; see Supplementary Fig. 2b for verification of recombination efficacy). The effect of the mutation was analyzed in vitro and in vivo, i.e., in stem cells associated with floating myofibers and in the regenerating muscle. On fibers cultured for 60 h, the associated colonies contained similar numbers of cells regardless of whether the fibers were obtained from *TxDll1^f/f* or control mice (Fig. 1c). However at 72 h, the colony size on fibers from mutant mice was reduced, which was accompanied by a large increase in the number of MyoG+ cells and fewer Pax7+ cells (Fig. 1c, d). We conclude that myogenic cells on fibers from *TxDll1^f/f* animals had a higher propensity to differentiate and to turn on MyoG. It should be noted that the majority of MyoG+ cells have exited the cell cycle, accounting for the reduction in the colony size. Since in this genetic experiment Dll1 is mutated in stem cells but not myofibers, the results unambiguously demonstrate that Dll1 produced by activated and differentiating stem cells suppresses differentiation of neighboring cells. This is reminiscent of the mechanism of lateral inhibition first described in invertebrates[44,45].

Next we tested the role of *Dll1* in the regenerating muscle using *TxDll1^f/f* animals (see Fig. 1e for an outline of the experiment). This mutation did not affect the maintenance of muscle stem cells (Supplementary Fig. 2c). Activation of muscle stem cells occurred correctly, as assessed by the quantification of Pax7+ cells and of EdU incorporation into Pax7+ cells of *TxDll1^f/f* mice at early stages of regeneration (3 days post injury, 3dpi; Supplementary Fig. 2d). However, at later stages the cells differentiated prematurely in vivo, and we observed decreased numbers of Pax7+ and increased numbers of MyoG+ cells at 4dpi (Fig. 1f). Accordingly, few Pax7+ cells remained in the regenerated muscle of *TxDll1^f/f* mice at 7dpi and 21 dpi (Fig. 1g). Analysis of newly formed fibers at 7 and 21 dpi demonstrated that these contained less nuclei in *TxDll1^f/f* than control mice. Further, the myofiber diameter was severely reduced at 21 dpi but little affected at 7 dpi (Fig. 1g). We conclude that muscle regeneration was severely impaired by the *Dll1* null mutation due to premature differentiation of muscle stem cells that interfered with their self-renewal. This resembles the effect of a developmental *Dll1* mutation that results in premature differentiation of progenitor cells[6,8].

**Dll1 expression oscillates in both, muscle progenitors and activated muscle stem cells.** Bioluminescence imaging can be used to monitor the expression of luciferase fusion proteins in single cells. Since luciferase imaging does not require external excitation, photodamage is prevented. This allows for the observation of myogenic cells over long periods without impairing

their survival or differentiation[25,46]. We used the *Dll1^luc* gene which encodes firefly luciferase fused to Dll1 to monitor dynamic Dll1 expression in primary muscle stem cells. Myofibers with the attached muscle stem cells were isolated from animals carrying the *Dll1^luc* gene. Dll1-luciferase bioluminescence was neither detected in muscle fibers nor in the associated quiescent stem cells. However, when fibers were cultured as floating fibers, we detected oscillating Dll1-luciferase bioluminescence in associated muscle stem cells, but never sustained expression (Fig. 2a and Supplementary Movie 1; the brightfield picture of the tracked cell is shown in Supplementary Fig. 3a). The oscillatory period was 2–3 h (Supplementary Fig. 3a). When fibers were cultured for 42 h prior to imaging, small colonies had formed in which the cells remain in contact with each other. Of note, when colonies contained two cells, we could unambiguously track the Dll1-luciferase dynamics in individual cells. Unfortunately, this was not possible in larger colonies because cells constantly changed their relative positions in the cluster. In two-cell colonies, we observed oscillatory bioluminescence in both cells, but never sustained expression (Fig. 2b and Supplementary Movie 2; see Supplementary Fig. 3b for a brightfield picture of the tracked cell). Again the oscillatory period observed was 2–3 h, and on average the oscillations in the two cells were out-of-phase with a mean phase shift corresponding to about half of an oscillatory period (Supplementary Fig. 3c). Therefore, in coupled cells, Dll1 is expressed in an oscillatory manner and each cell sends and receives Notch signals.

To assess the dynamics of Dll1 expression in groups containing more than two cells, we isolated muscle stem cells and cultured them as spheres. Spheres form spontaneously when the cells are cultured on a non-adhesive substrate. In the spheres, muscle stem cells constantly contacted each other. This cannot be achieved when the cells are plated on adhesive substrates due to their motility. Before sphere formation, the stem cells were co-transfected with plasmids encoding EpDll1-NanoLuc reporter and nGFP (Supplementary Fig. 3d). Transfected and non-transfected cells were mixed at a ratio of 1:50. Therefore, only a few cells in the sphere expressed nGFP and EpDll1-NanoLuc, which allowed to follow Dll1 expression dynamics. Bioluminescence imaging demonstrated that the NanoLuc reporter was expressed in an oscillatory manner in spheres formed by wild-type cells. The oscillatory period was 2–3 h (Fig. 2c; Supplementary Fig. 3e; Supplementary Movie 3).

Next, we also imaged myogenic progenitor cells during development using cultured slices from *Dll1^luc* mice at embryonic day (E) 11.5. These animals carried an additional *Pax7^nGFP* transgene to visualize myogenic progenitors. Again, we detected

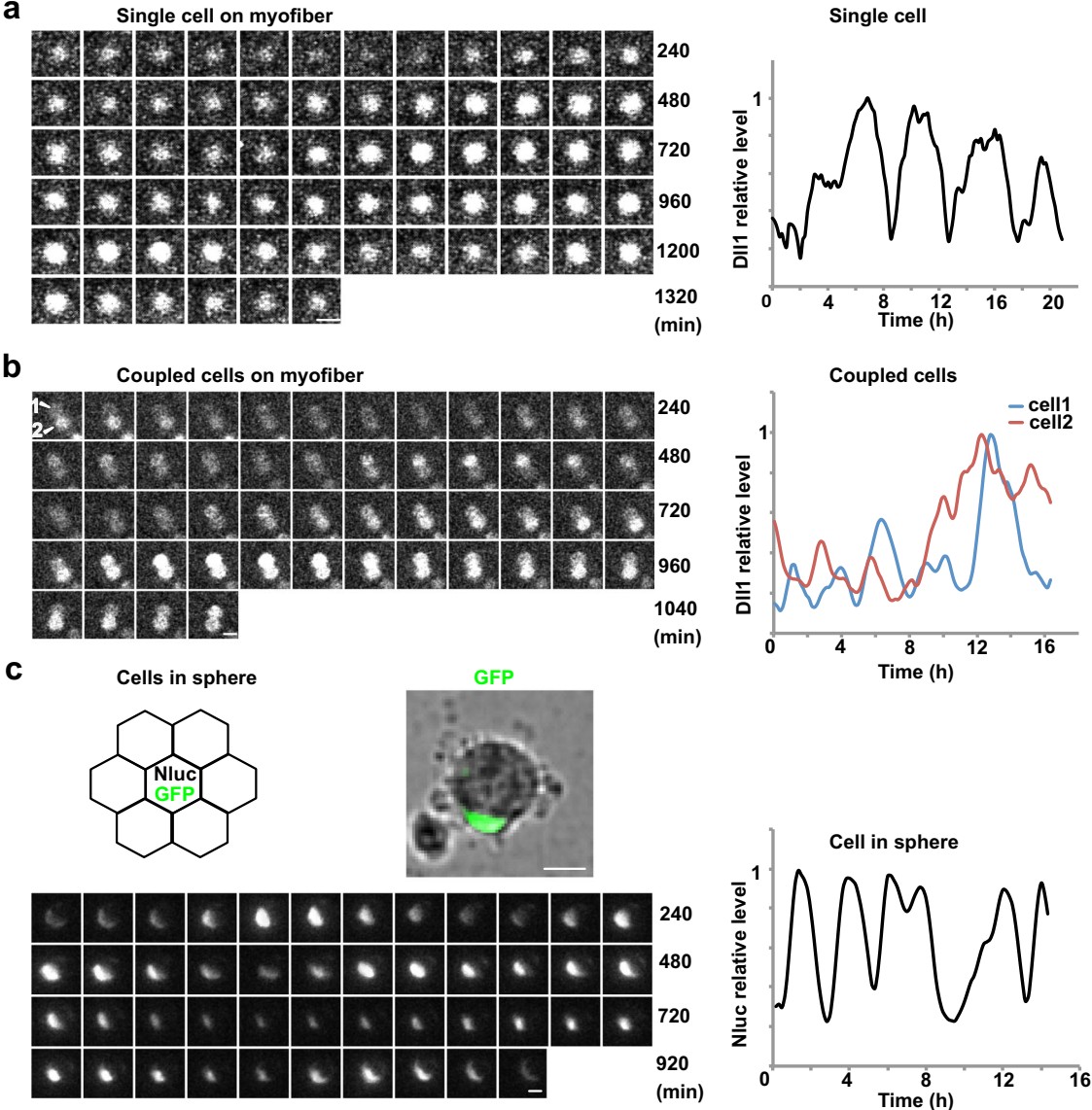

**Fig. 2 Dll1 protein expression oscillates in activated muscle stem cells. a** Bioluminescence images of Dll1-luciferase observed for an exemplary single muscle stem cell associated with a myofiber (left); imaging started after the fiber from a *Dll1*[luc] animal was incubated overnight. A quantification of the bioluminescence signal of this single cell is shown on the right. **b** Bioluminescence images of luciferase activity observed in two contacting muscle stem cells that are associated with a myofiber; the fiber was obtained from a *Dll1*[luc] animal. Imaging started after the fiber was incubated for 42 h. Quantifications of the bioluminescence signals observed in each of the two cells is shown on the right. **c** Schematic drawing of cells grown in a sphere; a single cell that is co-transfected with *EpDll1-NanoLuc* and *nGFP* expression plasmids is surrounded by many untransfected cells (upper left). NanoLuc bioluminescence signals detected in the single GFP+ muscle stem cell in a sphere (lower left); the cell is shown in the brightfield image above. Quantification of the NanoLuc bioluminescence signal detected in this cell (lower right). Scale bars, 15 µm.

oscillatory expression of Dll1 in myogenic cells, and an oscillatory period of 2–3 h (Supplementary Fig. 3f). We conclude that Dll1 expression oscillates in developing and adult muscle stem cells.

**The Hes1 oscillator drives oscillatory expression of Dll1, while MyoD regulates robust Dll1 expression levels.** The oscillatory period of Dll1 in myogenic cells is similar to the one of MyoD and Hes1 proteins described previously[25]. We therefore asked whether oscillatory MyoD and/or Hes1 expression drives oscillatory Dll1 expression. MyoD binding sites were previously identified by ChIP-seq analysis of myogenic cells[47], one of which is located inside the fourth intron of *Dll1* (enhancer fragment *EF* indicated in Fig. 3a). We used a dual reporter luciferase assay in HEK293 cells that do not endogenously express myogenic

transcription factors, and tested a 151 basepair (bp) fragment spanning the binding site for enhancer activity. This fragment contains 3 and 1 E- and N-box sequences, respectively. We observed a 48- and 16-fold increase in enhancer activity when a plasmid containing the *EF* construct was co-transfected with MyoD or MyoG expression plasmids, respectively (Fig. 3b). When a *Hes1* expression plasmid was co-transfected, basal expression as well as MyoD-induced enhancement were decreased. Thus, the *EF* sequence from the *Dll1* intron possesses enhancer activity and responds to MyoD, Hes1 and MyoG. Further, ChIP-PCR experiments demonstrated that endogenous Hes1 and MyoD bind to the *EF* fragment of the *Dll1* locus in myogenic C2C12 cells (Fig. 3c, d). A previously characterized MyoD binding site in the Myomaker (*Mymk*) gene, as well as

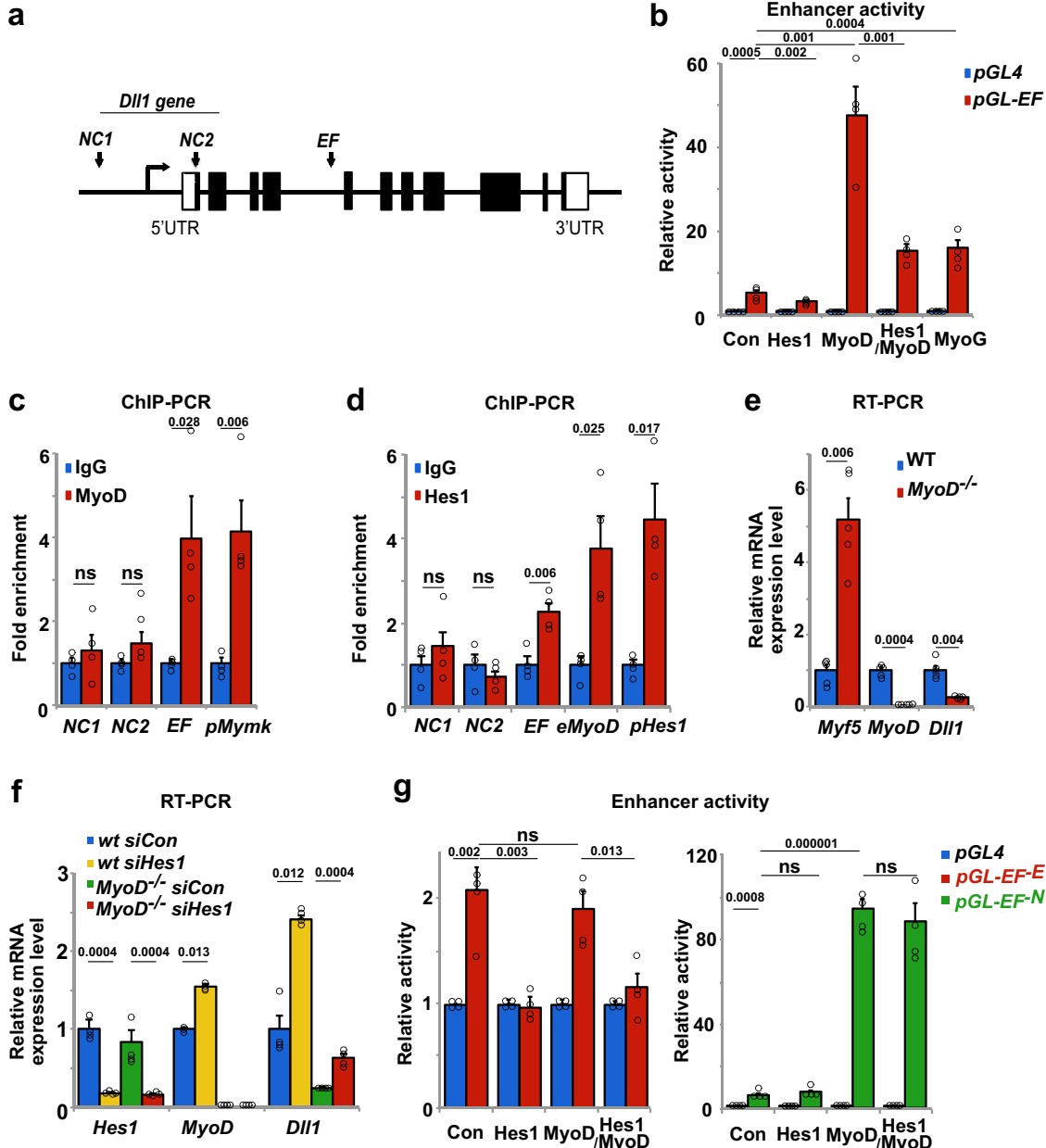

**Fig. 3 Expression of Dll1 is regulated by MyoD and Hes1. a** Schematic display of the *Dll1* gene; the enhancer fragment (*EF*) corresponds to sequences located in the fourth intron. Two additional fragments were used as negative controls (*NC1*, *NC2*). **b** Test of the enhancer activity of the *EF* fragment using the dual-luciferase reporter system (*pGL4* luciferase plasmid without an enhancer, blue bars; *pGL-EF* luciferase plasmid containing *EF* enhancer, red bars); cells co-transfected with *pCAG-nGFP* (control, con), *Hes1*, *MyoD*, *Hes1/MyoD* and *MyoG* expression plasmids were analyzed; *n* = 4 experiments. **c** ChIP-PCR experiment analyzing MyoD binding to *EF* and *NC1/NC2*. The known binding site in the *Myomaker* (*pMymk*) gene was used as a positive control; *n* = 4 experiments. **d** ChIP-PCR experiment analyzing Hes1 binding to *EF*, and to *NC1/NC2*. Known binding sites for Hes1 in the *MyoD* locus (*eMyoD* 23 kb upstream the *MyoD* transcript initiation site) and the *Hes1* promoter (*pHes1*) were used as positive controls; *n* = 4 experiments. **e** qPCR analysis of *Dll1*, *MyoD* and *Myf5* transcripts in wild-type (wt) and *MyoD⁻/⁻* mutant muscle stem cells, demonstrating MyoD-dependent *Dll1* expression; *n* = 5 animals. **f** qPCR analysis of *Hes1*, *MyoD* and *Dll1* in muscle stem cells isolated from wild-type and *MyoD⁻/⁻* mice; cells were further treated with control or *Hes1* siRNAs; *n* = 4 experiments. **g** Comparison of the enhancer activity using the dual-luciferase reporter system; *pGL4* luciferase plasmid without an enhancer (blue bars); *pGL4-EF⁻ᴱ* luciferase plasmid containing enhancer lacking E-box sequences (red bars); *pGL4-EF⁻ᴺ* luciferase plasmid containing enhancer lacking N-box sequences. Control (con) cells co-transfected with *pCAG-nGFP*, and cells co-transfected with *Hes1*, *MyoD*, and *Hes1/MyoD* expression plasmids were analyzed; *n* = 4 experiments. Data are presented as mean values ± SEM. Exact *p* values are indicated, unpaired two-sided *t*-test.

known Hes1 binding sites from the *MyoD* and *Hes1* loci were used as positive controls[25,48,49]. Two negative control sequences of the *Dll1* gene, *NC1*, and *NC2*, were not enriched in the ChIP experiments. Finally, we tested whether MyoD and Hes1 regulate *Dll1* expression in primary muscle stem cells. *Dll1* transcripts were reduced fourfold when freshly FAC-sorted muscle stem cells

from *MyoD⁻/⁻* and wild-type mice were compared. Of note, *Myf5* is upregulated in *MyoD⁻/⁻* cells, but this did not rescue *Dll1* expression (Fig. 3e). When we used siRNA knockdown of *Hes1*, *Dll1* transcript levels were increased around 2.5-fold in the presence or absence of *MyoD* (Fig. 3f; see Supplementary Fig. 4a for siRNA efficacy). Thus, MyoD and Hes1 enhance and repress

*Dll1* transcription, respectively, and directly bind to enhancer sequences in the *Dll1* gene.

We used two mutant *EF* fragments for dual reporter luciferase assays in HEK293 cells, one lacking all MyoD binding sites ($EF^{-E}$; CAGCTG replaced by CAGtTt), and a second lacking the Hes1 binding site ($EF^{-N}$; CACCAG replaced by CAaaAG) (Supplementary Fig. 4b). The $EF^{-E}$ sequence no longer enhanced transcription in the presence of MyoD, but still responded to Hes1. Conversely, the $EF^{-N}$ sequence no longer responded to Hes1, but MyoD still activated transcription driven by this fragment (Fig. 3g). This indicates that MyoD and Hes1 function independently of each other. Further, ChIP-PCR analysis showed that MyoD bound the wild-type and $EF^{-N}$ fragments, but not $EF^{-E}$, whereas Hes1 bound the wild-type and $EF^{-E}$ fragments, but not $EF^{-N}$ (Supplementary Fig. 4c). We conclude that MyoD and Hes1 bind and function independently of each other when controlling *Dll1* expression using the *EF* enhancer.

Next we experimentally tested whether MyoD is required for oscillatory Dll1 expression. Fibers were isolated from $MyoD^{-/-}$ animals that carried in addition a $Dll1^{luc}$ allele, and luciferase imaging of the associated activated muscle stem cells was performed. The Dll1-luciferase protein still oscillated in $MyoD^{-/-}$ cells (Fig. 4a; the brightfield image of the tracked cell is shown in Supplementary Fig. 5a). Quantifications using Fast Fourier transformation (power of FFT) showed that the stability of the Dll1 oscillations was not significantly affected, and the oscillatory period was also unchanged (Fig. 4a). However, luciferase levels were reduced compared to control muscle stem cells and we therefore needed longer exposure times to detect luciferase (9 and 6 min for *MyoD* mutant and control cells, respectively).

We used conditional mutagenesis and siRNA knockdown to experimentally test Hes1's effect on Dll1 oscillations. Dll1-luciferase produced by $Dll1^{luc}$ no longer oscillated in a stable manner in activated stem cells on cultured fibers obtained from mice with a *TxHes1* genetic background ($Pax7^{IRESCreERT2}$; $Hes1^{flox/flox}$ treated with tamoxifen). Instead, we observed sustained expression interrupted by small, irregular fluctuations of Dll1-luciferase (Fig. 4b and Supplementary Fig. 5b). The amplitudes of these fluctuations were typically smaller than the oscillatory amplitudes, and they display no periodicity. We therefore consider these small and irregular fluctuations as noise in gene expression. Such noise is commonly observed in genetically identical cells and was assigned to the stochasticity of processes participating in gene expression[50]. Similar to the mutation, treatment with *Hes1* siRNA interfered with Dll1-luciferase oscillations in stem cells associated with fibers, whereas control siRNA had no effect (Fig. 4c and Supplementary Fig. 5c, d). Thus, oscillatory Dll1 expression depends on Hes1. The presence of a *MyoD* mutation did not rescue oscillatory Dll1-luciferase expression after *Hes1* siRNA treatment (Fig. 4d and Supplementary Fig. 5e). Further, *Hes1* siRNA treatment resulted in increased differentiation propensity of cells in the fiber-associated colonies that was not observed in $MyoD^{-/-}$ cells treated with *Hes1* siRNA (Supplementary Fig. 5f), thus substantiating our previous work[6,25]. Together, these experiments show that the dynamic repression by Hes1 drives Dll1 oscillations, whereas MyoD affects the expression level but is not essential for Dll1 oscillations.

**Modeling of the oscillatory expression network in single and coupled myogenic cells.** The above analysis indicated that Notch components and myogenic differentiation factors participate in an oscillatory network in which the individual components control each other's expression (see Fig. 5a for a scheme). We used mathematical modeling to further describe the oscillatory network

(see Supplementary Methods for more information about the model). The first model relies on our experimental findings showing that Hes1 directly represses *Dll1* and *MyoD* and that MyoD directly enhances *Dll1* transcription (this work and ref.[25]). Further, it uses previously published parameters for *Dll1*, *Hes1*, and *MyoD* mRNA and protein stability, thus extending our previous model for Hes1 and MyoD oscillations in single cells[25,30,36,37]. As observed experimentally, the model predicted that all three proteins Hes1, MyoD, and Dll1 oscillate with similar periods (Fig. 5b). We used the model to test the effect of *Hes1* and *MyoD* ablation on Dll1 oscillations. In accordance with our experimental observations, ablation of *Hes1* was predicted to enhance *Dll1* expression and to interfere with oscillatory Dll1 expression, whereas *MyoD* ablation was predicted to affect expression levels but not Dll1 oscillations (Supplementary Fig. 6a, b). As MyoG+ cells do not co-express Hes1 (Supplementary Fig. 6c), Dll1 is predicted to be expressed in a sustained manner in MyoG+ cells.

We extended the mathematical framework, using a delay differential equation model to simulate the expression dynamics of single and coupled cells (Fig. 5c–d for two coupled cells, and Supplementary Fig. 6d–f for a single cell; see also Supplementary Methods for more information about the modeling approach). The extended model is based on our first single-cell model and further builds on a framework described previously that relied on experiments to estimate Hes1 dynamics. In particular $\tau_{21}$, the time that Hes1 requires to affect Dll1 protein levels, was determined to be 0.35 h in wild-type cells[36,51]. In agreement with our experimental observations, the model predicts that in two coupled cells Dll1 will oscillate in both cells, and that these oscillations will occur with a shift of half a phase period (Fig. 5d).

The value of $\tau_{21}$ includes the time needed for *Dll1* transcription that can be experimentally manipulated by changing the length of the primary transcript, for instance by the $Dll1^{type2}$ mutation (see also below). The model predicts that in a single cell an increase of $\tau_{21}$ by 0.1 h neither affects the oscillatory expression nor the oscillatory period of Dll1 (Supplementary Fig. 6d–f). However, in coupled cells, such an increased transcriptional delay is predicted to severely quench the entire oscillatory system (Fig. 5d; Supplementary Fig. 6g), which is in accordance with a simpler model developed previously[36]. Unexpectedly, the model also predicted that when only one of two coupled cells possesses a prolonged transcriptional delay, oscillations are only moderately quenched (Fig. 5d).

**Oscillatory Dll1 expression ensures the correct balance between differentiation and self-renewal of muscle stem cells in the regenerating muscle.** We used a previously generated Dll1 allele, $Dll1^{type2}$, in which the transcript length of *Dll1* was increased by the insertion of *Dll1-luciferase* cDNA into its first exon. The insertion results in an increase of the *Dll1* transcription time by 0.1 h[36] (see Fig. 6a for a scheme of $Dll1^{type2}$). The allele allows an analysis of the role of Dll1 oscillations in myogenesis and, at the same time, analysis of dynamic Dll1 expression by bioluminescence imaging. First, expression levels of *Dll1* as well as Notch target genes were compared in isolated muscle stem cells from control and $Dll1^{type2}$ mutant mice. Several factors of the Hes/Hey family are targets of Notch in muscle stem cells. Genetic data indicated that among these Hes1 is the functionally dominant factor[25]. qPCR analysis demonstrated that the expression of *Dll1*, *Hes1*, *Hey1* and *Hes5* were unchanged in the regenerating muscle stem cells (3 dpi) and developing muscle progenitors (E12.5) (Supplementary Fig. 7a). The developmental phenotypes observed in homozygous $Dll1^{type2}$ mice preclude their analysis in adulthood[36], and we used conditional mutagenesis to generate adult $Dll1^{type2}$ mutant muscle stem cells (muscle stem cells

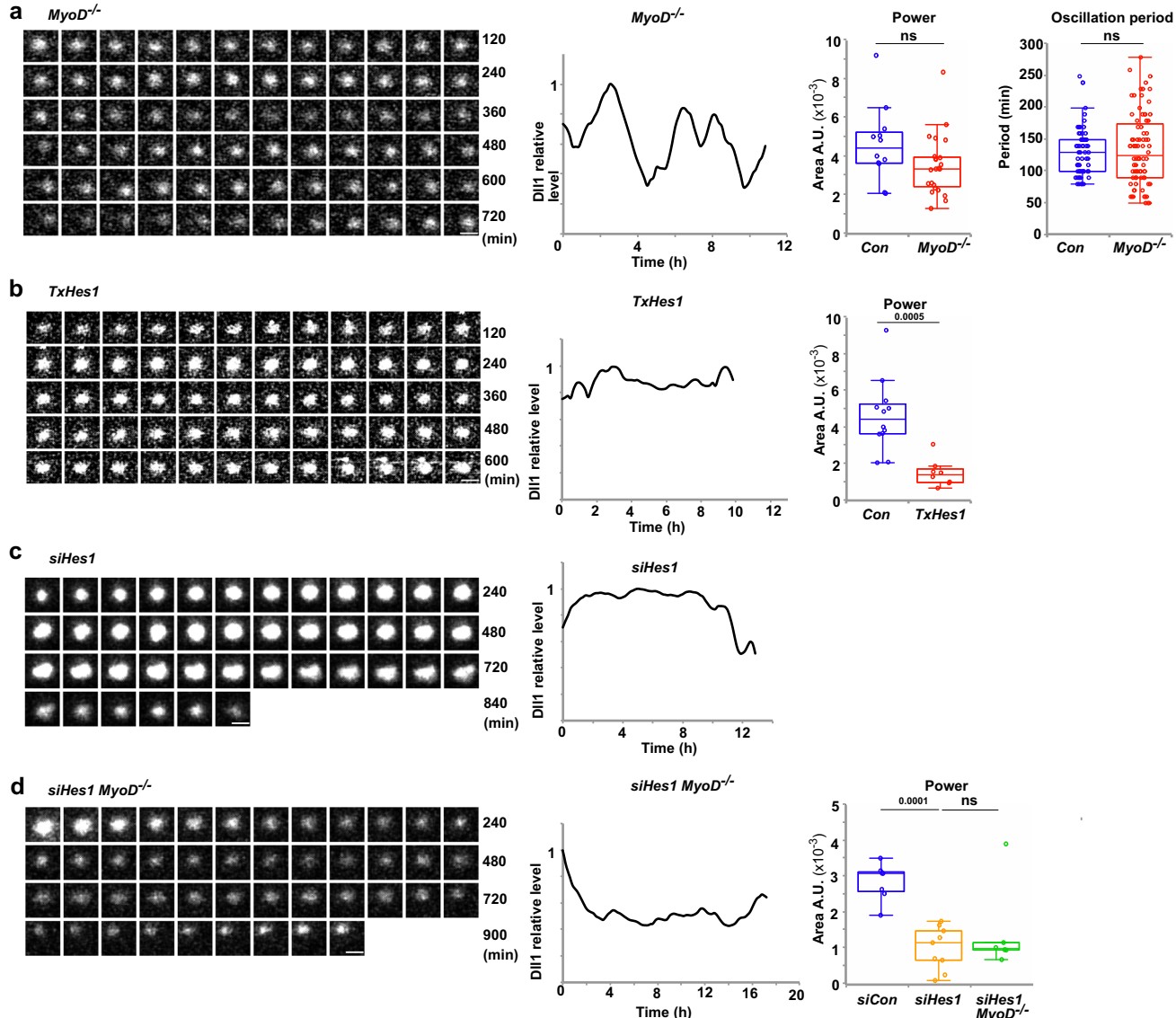

**Fig. 4 Hes1 but not MyoD is required for the oscillatory expression of Dll1. a** Bioluminescence signals from Dll1-luciferase observed in a single *MyoD*−/− mutant muscle stem cell associated with a myofiber and a quantification of this bioluminescence signal (left); the myofiber and associated stem cell was isolated from a *MyoD*−/−;*Dll1*luc animal. Quantification of the oscillatory stability (power of the Fast Fourier transformation) and the oscillatory period in control and *MyoD*−/− cells (right); each point represents data obtained from a single imaged cell; *n* = 23 experiments. **b** Bioluminescence signals from Dll1-luciferase observed in a single *Hes1* mutant muscle stem cell associated with a myofiber obtained from a *TxHes1*;*Dll1*luc animal and quantification of this bioluminescence signal (left). Quantification of the oscillatory stability (power of the Fast Fourier transformation) in control and *Hes1* mutant cells (right); *n* = 8 experiments. **c** Bioluminescence signals from Dll1-luciferase observed in a single muscle stem cell associated with a myofiber treated with siHes1 RNA and quantification of this bioluminescence signal. The myofiber and associated muscle stem cell were isolated from a *Dll1*luc animal. **d** Bioluminescence signals from Dll1-luciferase observed in a single muscle stem cell associated with a fiber and quantification of this bioluminescence signal (left); the fiber and associated stem cell were derived from a *MyoD*−/−;*Dll1*luc animal and treated with *siHes1* RNA. Quantification of the oscillatory stability (power of the Fast Fourier transformation) in siRNA control (*n* = 5 experiments) and *siHes1* RNA (*n* = 9 experiments) treated wild-type cells, and *siHes1* RNA treated *MyoD*−/− (*n* = 6 experiments) mutant cells (right). In the box plot, center lines show the medians; box limits indicate the 25th and 75th percentiles; whiskers extend 1.5 times the interquartile range. Scale bars, 15 μm. Exact *p* values are indicated, ns indicates *P* > 0.05, unpaired two-sided *t*-test.

obtained from *Pax7*IRESCreERT2;*Dll1*type2/flox mice treated with tamoxifen, named hereafter *TxDll1*f/type2 animals). Single activated muscle stem cells from such mutants displayed oscillatory luciferase expression on fibers after 24 h of culture (Fig. 6b; Supplementary Movie 4, Supplementary Fig. 7b). When we imaged two coupled cells in colonies that had formed after 42 h of culture, stable oscillations were undetectable. Instead luciferase displayed sustained expression interrupted by small, irregular fluctuations (Fig. 6c; Supplementary Movie 5; Supplementary Fig. 7c). We conclude that the *Dll1*type2 mutation in muscle stem

cells disrupts Dll1 oscillations in coupled but not in single activated muscle stem cells.

We isolated *Dll1*type2 mutant stem cells from *TxDll1*f/type2 animals, transfected them with the *EpDll1-NanoLuc* reporter plasmid, and cultured them as spheres. Bioluminescence imaging demonstrated that expression of the NanoLuc reporter did not oscillate (Fig. 6d; Supplementary Fig. 7d; Supplementary Movie 6), substantiating the observation on coupled cells on fibers. Of note, we can distinguish signals from the EpDll1-NanoLuc protein and the luciferase present in the *Dll1*type2 mutant allele because

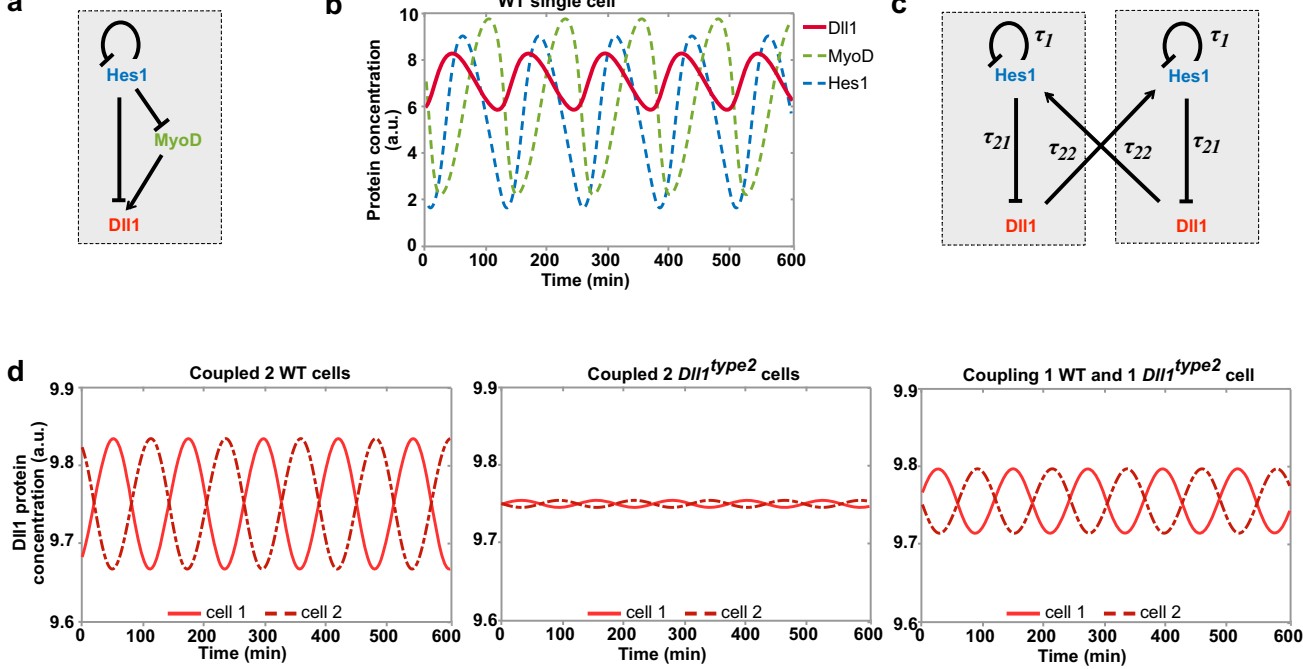

**Fig. 5 Mathematical modeling of the dynamic Dll1 expression in single and coupled wild-type and *Dll1type2* mutant muscle stem cells. a** Scheme of the transcriptional regulation of *Dll1* by Hes1 and MyoD. **b** Prediction of the expression dynamics of Dll1, MyoD, and Hes1 proteins using a single-cell ordinary differential equation model. **c** Scheme of the Dll1 and Hes1 regulatory mechanisms in two coupled cells underlying the coupled-cell delay differential equation model. In each cell, Hes1 represses its own as well as *Dll1* transcription in a cell-autonomous manner; in addition, Dll1 in one cell induces Hes1 in the neighboring cell. **d** Simulation of the dynamic expression of Dll1 in two coupled wild-type (WT) cells (left), two coupled *Dll1type2* mutant cells (middle), and a chimeric situation in which one cell is *Dll1type2* mutant and the other is wild-type (WT). See Methods and Supplementary Methods for detailed information about the derivation and parametrization of the mathematical models.

NanoLuc and firefly luciferase use distinct substrates. We next tested chimeric situations, i.e., *Dll1type2* mutant stem cells transfected with the *EpDll1-NanoLuc* reporter that were mixed 1:50 with wild-type cells. In such chimeric spheres, the surrounding wild-type cells allowed an oscillatory Dll1 expression in the *Dll1type2* mutant stem cell. Similarly, when wild-type cells were surrounded by *Dll1type2* mutant cells, NanoLuc oscillated in the wild-type cells (Supplementary Fig. 7e–g). In summary, the *Dll1type2* mutation interferes with stable oscillations in 2-cell colonies or in large cell groups in which myogenic stem cells contact each other. In contrast, *Dll1type2* mutant cells still express Dll1 in an oscillatory manner when they are mixed with wild-type cells in spheres.

We used *Dll1type2/type2* mice to visualize the dynamic expression of Dll1 in myogenic cells during development, i.e., cultured slices from limbs of E11.5 mice. Luciferase expression no longer oscillated and instead we observed sustained expression interrupted by small and irregular luciferase fluctuations. FFT was used to quantify the luciferase expression dynamics in cultured slices from *Dll1type2/type2* embryos, which also demonstrated that stable oscillations were no longer observable (Supplementary Fig. 7h).

We assessed the functional consequences of quenched Dll1 oscillations in vitro (cultured floating fibers, sphere cultures) or in vivo (regenerating muscle). After 60 h in culture, colonies on the fibers contained similar numbers of cells regardless of whether the fibers were obtained from *TxDll1f/type2* or control mice. However after 72 h in culture, the colony size on mutant fibers was reduced, which was accompanied by a large increase in the number of MyoG+ cells and a decrease in the number of Pax7+ cells (Fig. 7a, b). In sphere cultures, *Dll1type2* mutant muscle stem cells showed a higher propensity to differentiate than wild-type

cells as assessed by determining the percentage of MyoG+ and Pax7+ cells (Fig. 7c). However, when *Dll1type2* mutant cells were surrounded by wild-type cells in chimeric spheres, differentiation was suppressed (Supplementary Fig. 8a), which is in accordance with the rescue of the oscillatory behavior observed in these chimeric conditions (Supplementary Fig. 7f, g). In summary, these results show that not only the presence of Dll1 but also its expression dynamics determines the balance between self-renewal and differentiation of coupled muscle stem cells in culture.

Next we assessed the consequences of the *Dll1type2* mutation in vivo. The ratio of MyoG+ to Pax7+ cells in the regenerating muscle of *TxDll1f/type2* animals was strongly increased compared to control mice at early stages of regeneration (Fig. 7d). Nevertheless, the proliferative capacities of Pax7+ cells were unchanged (Supplementary Fig. 8b). Interestingly, while average Hes1 protein levels were similar in Pax7+ cells of control and mutant muscle, the Hes1 variance was larger in control than *Dll1type2* mutant cells (Fig. 7d), indicating that Hes1 oscillations were also affected. We further analyzed the variance of Hes1 protein by distinguishing between single and coupled myogenic cells. Thus, Pax7+ cells that were or were not directly contacting other myogenic cells were identified in the regenerating muscle, and their Hes1 expression levels were compared. This demonstrated that Hes1 variance was similar in single Pax7+ cells, but distinct in coupled Pax7+ cells from control and *TxDll1f/type2* mice, further supporting the notion that the effect of the mutation depends on cell coupling (Fig. 7d). At later stages of regeneration (7 and 21 dpi), a severe depletion of Pax7+ cells was observed in *TxDll1f/type2* mutants (Fig. 7e). Newly formed fibers were aberrant, containing fewer nuclei at 7 and 21 dpi; fiber diameter was significantly smaller at 21 but not 7dpi (Fig. 7e). We conclude that the balance between self-renewal and differentiation of

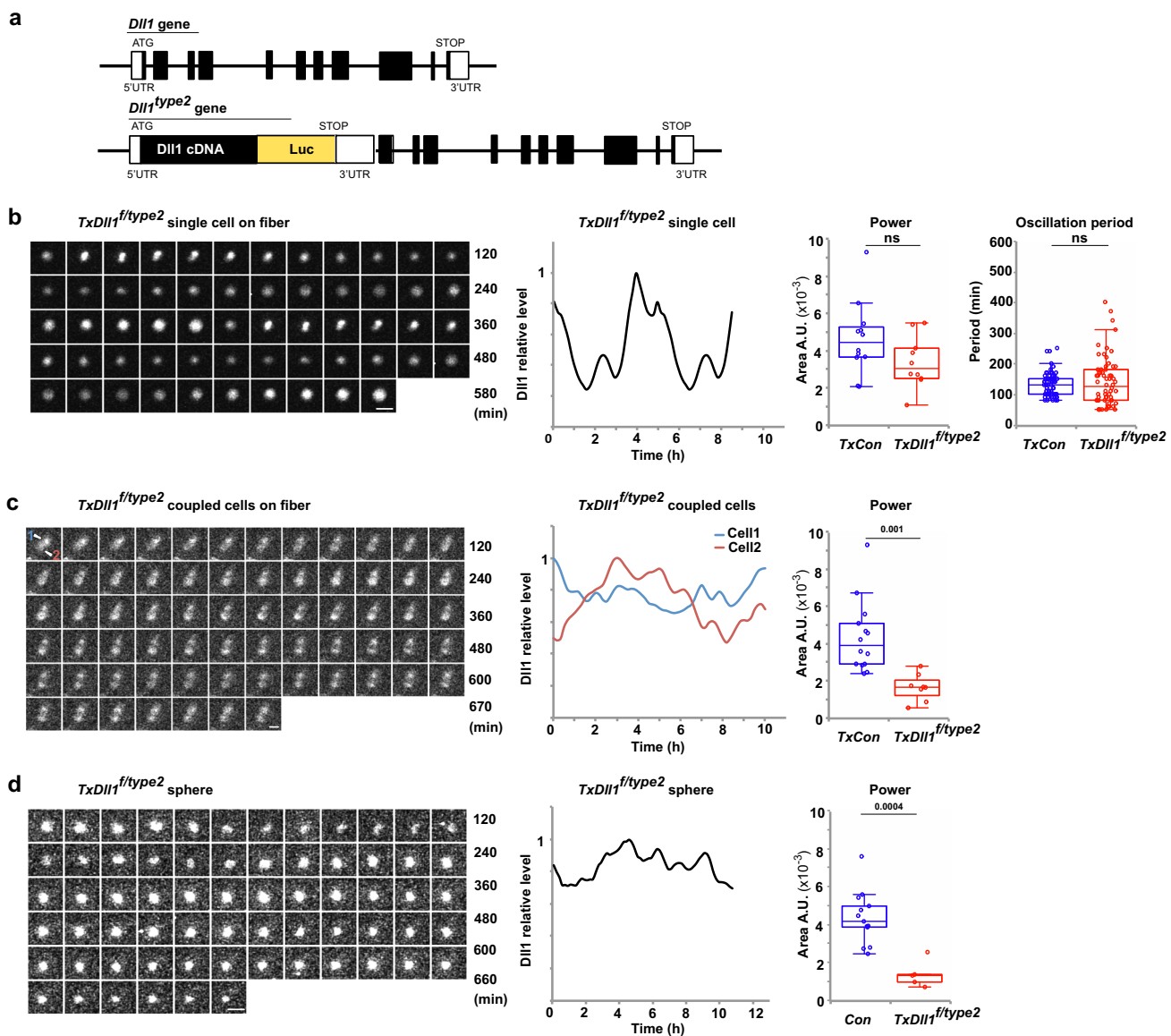

**Fig. 6 The *Dll1type2* mutation interferes with oscillatory Dll1 expression. a** Schematic display of *Dll1* gene and the *Dll1type2* mutant allele; in *Dll1type2* a fused cDNA encoding Dll1 (black) and firefly luciferase (luc, yellow) were inserted into the *Dll1* locus; the 5′ and 3′ UTR, a translational stop codon (Stop), and the initiation codon (ATG) are indicated. **b** Bioluminescence images observed in a single *Dll1type2* mutant muscle stem cell associated with a myofiber and quantification of this bioluminescence signal (left); the fiber and associated stem cell were obtained from a *TxDll1f/type2* animal. Quantification of the oscillatory stability (power of the Fast Fourier transformation) and the oscillatory period of luciferase bioluminescence in control and *Dll1type2* mutant cells (right); *n* = 10 experiments. **c** Bioluminescence images observed in two *Dll1type2* mutant muscle stem cells contacting each other on a cultured myofiber and quantification of the bioluminescence signals in each of the two cells (left; signals from cell 1 and 2 are shown in blue and red, respectively). Quantification of the oscillatory stability (power of the Fast Fourier transformation) of luciferase bioluminescence in coupled control and *Dll1type2* mutant cells (right); *n* = 4 experiments. **d** NanoLuc bioluminescence signals and quantification of these signals in a cell located in a sphere of *Dll1type2* mutant cells (left); one cell co-transfected with an *nGFP* and *EpDll1-NanoLuc* expression plasmid was monitored. Quantification of the oscillatory stability (power of the Fast Fourier transformation) in sphere cultures of *Dll1type2* mutant cells (right); *n* = 5 experiments. In the box plot, center lines show the medians; box limits indicate the 25th and 75th percentiles; whiskers extend 1.5 times the interquartile range. Scale bars, 15 μm. Exact *p* values are indicated, ns indicates *P* > 0.05, unpaired two-sided *t*-test.

coupled muscle stem cells during muscle regeneration depends on the oscillatory input of the Dll1 signal, and that the mere presence of Dll1 does not suffice to ensure the correct balance.

Finally, we analyzed the effect of the *Dll1type2* mutation on developmental myogenesis. This demonstrated that dampened oscillation of Dll1 had severe consequences on muscle growth. At E14.5, a higher ratio of MyoG+/Pax7+ cells was apparent in *Dll1type2/type2* compared to control animals. Nevertheless, Pax7+ cells were unaffected in their proliferative capacities (Fig. 8a). At

E17.5, the size of muscle groups was decreased, and the number of Pax7+ cells was very severely decreased (Fig. 8b). Thus, compared to control animals, Pax7+ progenitor cells in *Dll1type2/type2* mutants were more likely to progress to terminal differentiation. In summary, when we experimentally interfered with Dll1 oscillations using the *Dll1type2* mutation, myogenic progenitor cells had a higher propensity to differentiate during muscle development and regeneration. This occurred despite the fact that Dll1 expression levels were not affected.

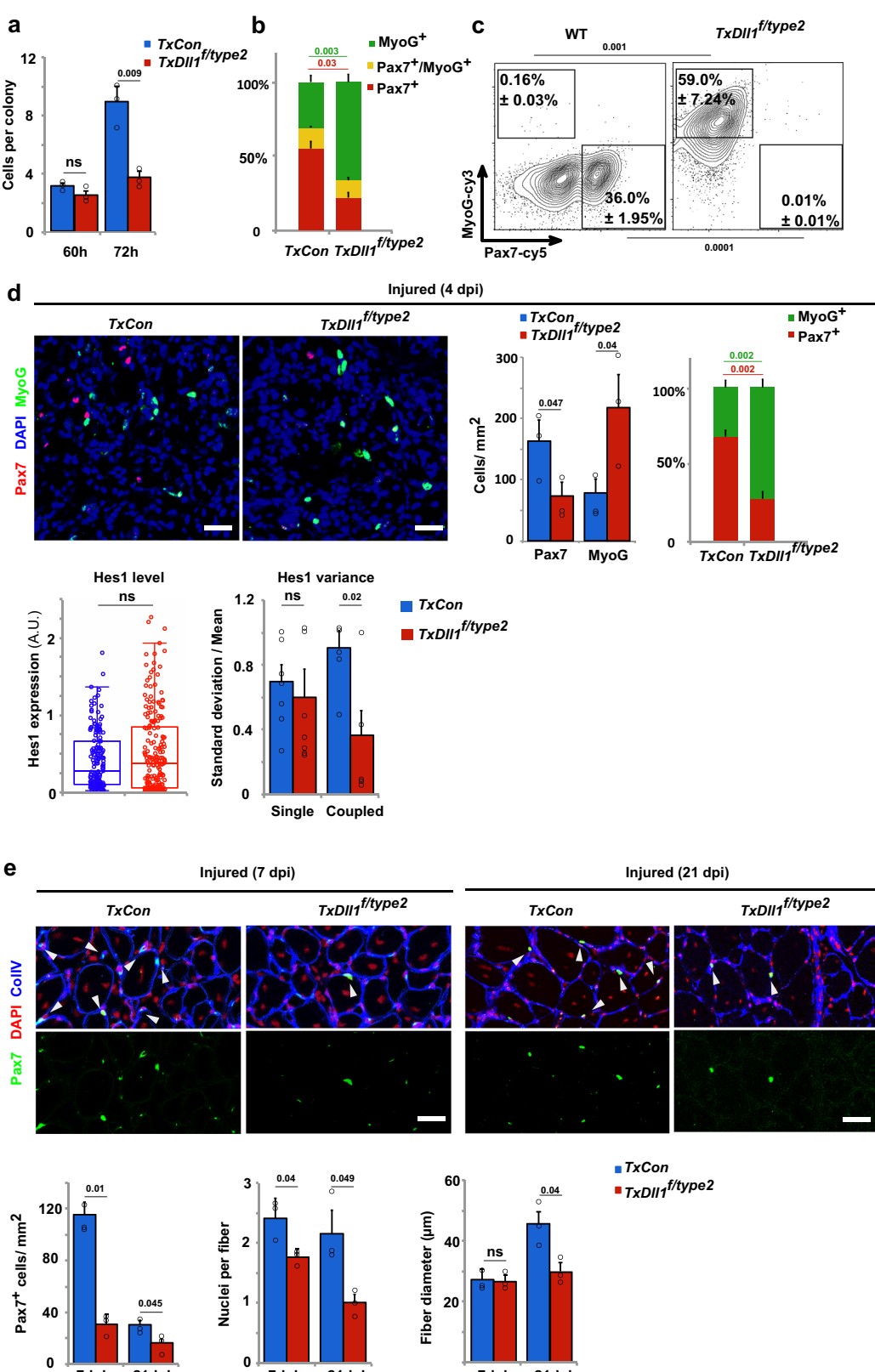

## Discussion

Notch-mediated lateral inhibition, a mechanism first described in invertebrates[44,45], is tuned in such a manner that it allows for the differentiation of a subpopulation of cells in a community, while simultaneously maintaining a stem cell pool. This is achieved because differentiating cells present a Notch ligand that represses the differentiation of neighboring cells. Here we report that communities of myogenic stem cells in development and regeneration rely on oscillatory Dll1 provided by neighboring cells that are in a proliferating state and express both, MyoD and Hes1[25].

**Fig. 7 Oscillatory Dll1 expression controls muscle regeneration. a** Quantification of the number of cells in colonies formed on myofibers; myofibers were isolated from *TxCon* (blue bars) and *TxDll1f/type2* mice (red bars) and cultured for 60 and 72 h; *n* = 3 animals. **b** Quantification of cells that express MyoG+ only (green), MyoG+ and Pax7+ (yellow), and Pax7+ only (red) in colonies associated with myofibers after 72 h of culture; fibers were isolated from *TxCon* and *TxDll1f/type2* mice; *n* = 3 experiments. **c** Gating strategy and representative FACS plot used to define MyoG+ (MyoG-cy3) and Pax7+ (Pax7-cy5) cells in cultured spheres containing wild-type (WT) and *Dll1type2* mutant cells; *n* = 3 experiments. **d** Immunohistological analysis of the regenerating muscle of *TxCon* and *TxDll1f/type2* mutant mice at 4 dpi using anti-Pax7 (red) and anti-MyoG (green) antibodies; DAPI (blue) was used as counterstain (upper left). Quantifications of the number of Pax7+ and MyoG+ cells in *TxCon* (blue bars) and *TxDll1f/type2* (red bars) muscle, and relative proportion of MyoG+ (green) and Pax7+ (red) cells in *TxCon* and *TxDll1f/type2* muscle (upper right); quantification of Hes1 expression levels in Pax7+ cells and variance of Hes1 protein levels in single and coupled Pax7+ cells of the regenerating muscle (4 dpi) of *TxCon* (blue bars) and *TxDll1f/type2* (red bars) animals (lower panels); *n* = 3 animals. **e** Immunofluorescence analysis of *TxCon* and *TxDll1f/type2* mutant mice at 7 and 21 dpi using anti-Pax7 (green) and anti-collagen IV (ColIV; blue) antibodies; DAPI (red) was used as counterstain (upper panels). Quantification of the number of Pax7+ cells, number of nuclei/fiber and fiber diameter in the regenerating muscle of *TxCon* (blue bars) and *TxDll1f/type2* (red bars) mice at 7 and 21 dpi (lower panels); *n* = 3 animals. Scale bars, 50 µm. Data are presented as mean values ± SEM. Exact *p* values are indicated, ns indicates *P* > 0.05, unpaired two-sided *t*-test.

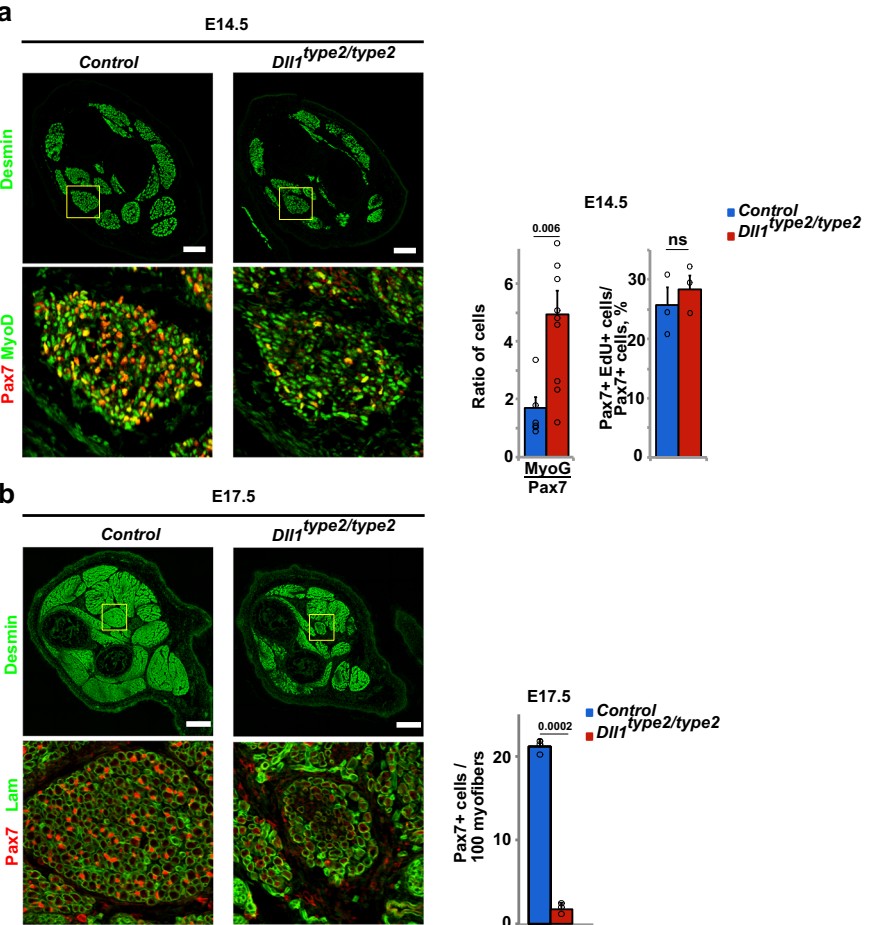

**Fig. 8 Oscillatory Dll1 expression controls muscle growth in fetal development. a** Immunohistological analysis of distal limb muscles of control and *Dll1type2/type2* mutant animals at E14.5 using the indicated antibodies. The ratio of MyoG+/Pax7+ cells and the quantification of the proliferation of Pax7+ cells (EdU incorporation into Pax7+ cells) is shown to the right (*n* = 3 animals). **b** Immunohistological analysis of distal limb muscles of control and *Dll1type2/type2* mutant animals at E17.5 using the indicated antibodies. Quantification of the number of Pax7+ cells in the muscle is shown at the right (*n* = 3 animals). Scale bars, 100 µm (**a**) and 200 µm (**b**). Data are presented as mean values ± SEM. Exact *p* values are indicated, ns indicates *P* > 0.05, unpaired two-sided *t*-test.

Dll1 oscillations result in a temporally dynamic situation where myogenic cells send and receive Notch input. Mutations that change the Dll1 oscillatory dynamics but not the Dll1 expression level result in premature differentiation of myogenic cells and severely affect muscle development and repair. Thus, Dll1 provided by a myogenic cell suppresses differentiation of neighboring myogenic cells, and Dll1 needs to oscillate to achieve the correct balance between self-renewal and differentiation (see summary in Fig. 9).

**Regulation of the oscillatory network**. Our previous work indicated that myogenic cells remain in a proliferative state as long as MyoD is expressed in an oscillatory manner and that they differentiate when MyoD expression is sustained. We also found that oscillatory expression of MyoD is controlled by oscillatory Hes1 and thus ultimately by Notch signaling[25]. Our new data presented here demonstrate that Dll1 expression also oscillates with a similar period as Hes1 and MyoD. This suggests that all three genes, *Hes1*, *MyoD*, and *Dll1*, are co-regulated in one

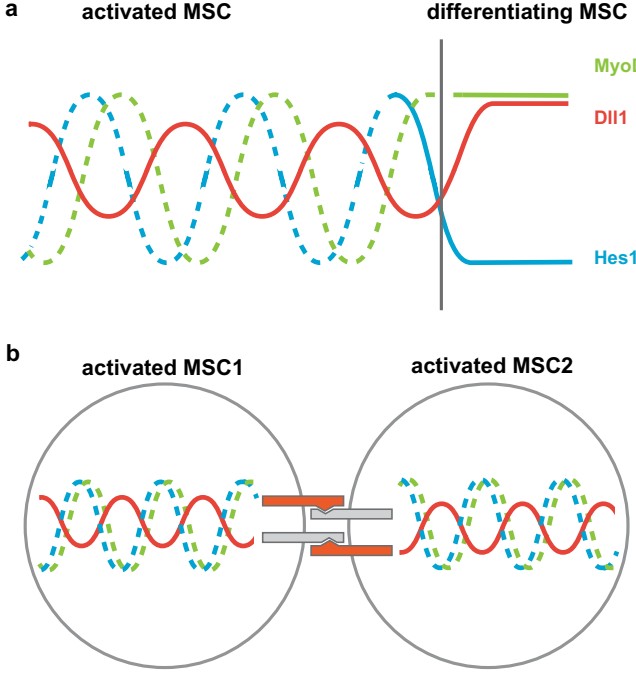

**Fig. 9 Oscillatory expression of Dll1, Hes1, and MyoD controls self-renewal and the timing of differentiation. a** Dll1, Hes1, and MyoD are dynamically expressed in activated muscle stem cells (left). Hes1 is no longer expressed, and Dll1 and MyoD expression are sustained when muscle stem cells differentiate (right). **b** In contacting cells that are activated, the periodic Dll1 input into the Notch signaling cascade needs to be coordinated between the two cells in order to allow stable oscillations. MSC muscle stem cell.

transcriptional network. A similar mechanism was previously suggested to coordinate the expression of the Notch signaling network and proneuronal genes[52,53]. We identified an enhancer of *Dll1* in myogenic cells that is controlled by both, Hes1 and MyoD, in a negative and positive manner, respectively. Our mathematical model and experimental data show that MyoD does not have a major impact on the oscillatory expression of Dll1 but regulates the Dll1 expression level. Contrastingly, ablation/downregulation of *Hes1* abolishes oscillations. Thus, Hes1 is the major oscillator in the myogenic system that drives the dynamic transcription of the network. Superimposed on this is the regulation of Dll1 expression levels by Hes1 and MyoD.

**Oscillatory versus stable Dll1 expression and the timing of myogenic differentiation.** Here, we used a *Dll1^type2^* mutant allele that interferes with Dll1 oscillations, but leaves the *Dll1* coding sequence and its expression levels unaffected. The sustained Dll1 expression in *Dll1^type2^* mutant cells was able to activate Notch signaling, as evident by the similar levels of direct Notch target genes in myogenic cells isolated from *Dll1^type2^* and control mice. Nevertheless, the *Dll1^type2^* mutation accelerated the timing of myogenic differentiation and precluded appropriate self-renewal during development and regeneration. This result demonstrates that the oscillatory Dll1 expression is functionally important for myogenesis. However, a null mutation of *Dll1* results in even more severe phenotypes than the *Dll1^type2^* mutation (compare Figs. 1g and 7e). Thus, oscillating Dll1 suppressed myogenic differentiation more efficiently than sustained Dll1, but sustained Dll1 produced in *Dll1^type2^* mutants retained a partial functionality.

Our observations show that in communities of myogenic cells, i.e., when cells receive and provide signals to one another, Dll1 oscillations drive the oscillatory network. Mathematical modeling shows that oscillators require in general specific coupling delays, and that inappropriate delays cause quenching, a phenomenon known as amplitude/oscillation death[54]. The *Dll1^type2^* mutation changes the delay time between initiation of *Dll1* transcription and protein production[36]. Our modeling of Dll1/Notch signaling between coupled cells indicated that an inappropriate delay of Dll1 expression dampens the oscillatory dynamics, providing the basis for the phenotypes observed in *Dll1^type2^* mutant mice, which is in accordance with earlier modeling data[36]. Thus, in communities of cells that receive and provide Notch signals, the appropriate timing of Dll1 input is important to maintain stable oscillations and to ensure an appropriate balance between self-renewal and differentiation. The *Dll1^type2^* mutation does not affect oscillatory Dll1 expression in single mutant muscle stem cells. However, the *Dll1^type2^* mutation interferes with the oscillatory expression of Dll1 in cellular communities that send and receive Notch signals, e.g., in myogenic colonies cultured on floating fibers or in spheres, as well as in the developing and regenerating muscle, i.e., in transit-amplifying cells that intermingle and contact each other.

Mathematical modeling predicts that stable Hes1 oscillations depend on the synthesis rates and on the half-lives of Hes1 protein and *Hes1* mRNA. Hes1 expression levels and Hes1 oscillations are controlled by Notch signaling but also by other stimuli like serum[55]. Further work is required to identify the stimuli that initiate and end Hes1 oscillations and thus the oscillations of the entire network in muscle stem cells, i.e. in cells that exit/enter quiescence or enter into terminal differentiation, respectively.

**Notch signals in developmental and regenerative myogenesis.** The Notch signaling pathway orchestrates quiescence as well as self-renewal in myogenic stem cells[7,23,25,43,56,57]. It is possible that distinct ligands and/or ligand sources act on these stem cells and control the two processes. By restricting the mutation to muscle stem cells and analyzing their behavior on cultured fibers, we show here unambiguously that oscillatory Dll1 produced by activated stem cells controls self-renewal of neighboring stem cells. In the in vivo setting of the regenerating muscle, stem cells form new fibers that potentially could also act as a Dll1 source. Our genetic experiments in the regenerating muscle do not exclude that Dll1 provided by myofibers might participate in the control of stem cell behavior. However, we neither detected *Dll1* transcripts in fibers using smFISH nor did we observe fiber-derived Dll1-luciferase signals using bioluminescence imaging or antibodies, arguing against such a mechanism. Myofibers and endothelia produce the Notch ligand Dll4, and Dll4 was proposed to control quiescence of myogenic stem cells[27,56,58,59]. Further, upregulation of *Mindbomb-1* (*Mib1*) in the myofiber which enhances Notch signaling in the muscle stem cell was recently implicated to control entry into quiescence[27,56]. In this context, it is interesting to note that Dll1 and Dll4 are functionally non-equivalent, and cell culture experiments using synthetic biological Notch networks indicate that Dll1 and Dll4 elicit pulsed and sustained responses in Notch signal-receiving cells, respectively[60,61]. Thus, differences in Notch signaling dynamics or, alternatively, distinct levels of the Notch signal might underlie quiescence and self-renewal responses in stem cells.

## Methods
**Bioluminescence imaging.** To analyze dynamic Dll1 expression in adult and embryonic muscle stem cells in wild-type, *MyoD^−/−^* and *TxHes1* genetic backgrounds, the *Dll1^luc^* allele was used. For analysis of dynamic Dll1 expression in

*Dll1type2* mutant cells or embryos, luciferase produced by the *Dll1type2* allele was imaged. For analysis of dynamic Dll1 expression in spheres, expression of Nanoluc produced by the *EpDll1-NanoLuc* indicator plasmid was used.

For imaging, myofibers were incubated in 35-mm glass-bottom dishes at 37 °C in 5% CO$_2$, and 1 mM luciferin was added to the culture medium immediately before imaging. For NanoLuc imaging, 100× diluted Endurazine (Promega, Wisconsin, USA) was added to the medium. Bioluminescence images were acquired by an inverted microscope (IX83-ZDC, Olympus, Tokyo, Japan) with a cooled EM-CCD camera (EM-X2 C9100-23B, Hamamatsu, Shizuoka, Japan) in a dark room. The filters and camera control were adjusted automatically using the CelSens software (Olympus, Tokyo, Japan). Frames were acquired with exposure times that were adjusted to the expression levels, i.e., 6–9 min exposure time for the luminescence signals[25,62].

**Single myofiber isolation, RNA interference**. Extensor digitorum longus (EDL) muscles were dissected, digested with 0.2% Collagenase I (Sigma-Aldrich, St. Louis, USA) for 1.5 h, and triturated using glass pipettes pre-coated with 5% BSA/PBS[25]. Single myofibers were picked and transferred to the medium (DMEM, 10% horse serum, 0.5% chicken embryo extract). For siRNA transfections, a complex of 50 nM siRNAs and Lipofectamin RNAiMax (ThermoFisher, Massachusetts, USA) was added to the medium after 4 h culture[63]. Myofibers were used for imaging or for immunostaining after the indicated culture times.

**Immunohistochemistry**. Immunohistology of slices was performed on 12 µm cryosections[9]. For adult tibialis anterior (TA) muscle, sections of tissue were fixed in PBS containing 4% paraformaldehyde for 20 min. For embryos, fixation at 4 °C with PBS containing 4% paraformaldehyde for 2 h was followed by overnight incubation with 20% sucrose in PBS for cryo-protection. Myofibers were fixed in PBS containing 4% paraformaldehyde for 10 min, followed by blocking with PBS containing 1% BSA for 1 h at room temperature. Sections or fibers were incubated with the primary antibodies overnight at 4 °C, washed in PBS, and incubated with secondary antibodies at room temperature for 1 h. The following antibodies were used: guinea pig anti-Pax7, rabbit anti-MyoD, mouse anti-MyoD, goat anti-desmin, goat anti-collagen IV, rabbit and mouse anti-MyoG, mouse anti-luciferase, and secondary antibodies conjugated with Cy2, Cy3, or Cy5. 4′,6-Diamidino-2-phenylindole (DAPI) was used as a nuclear counterstain; the source of the antibodies is listed in Supplementary Table 1. Images were acquired with a LSM700 confocal microscope (Zeiss, Jena, Germany).

**Single-molecule fluorescence in situ hybridization (smFISH)**. In situ hybridization was performed using the RNAscope Multiplex fluorescent V2 assay kit as recommended by the producer (ACD Biotech, CA, USA). Briefly, the assay allows simultaneous visualization of up to three mRNAs, with each probe being labeled by a distinct fluorophore and visualized on a different channel (C1-mDll1, C2-mPax7, C3-mMyoD or C3-mMyoG; see Supplementary Table 1 for more information). Fresh frozen sections and fixed frozen sections were used for hybridization.

**Cell and sphere culture**. Muscle stem cells were isolated using FACS[25]. For this, the dissected muscle was minced with scissors and digested with 14 mU/ml Collagenase (Serva, Halle, Germany) and 2.5 U/ml Dispase II (Roche, Basel, Switzerland) at 37 °C. The samples were filtered with 40 µm strainer (BD, New Jersey, US). Sca1−/CD31−/CD45−/Vcam+ cells were isolated by FACS. The cells were transferred on a glass-bottom dish coated with 10% Matrigel; after 20 min, growth medium (DMEM/F12, 15%FBS, supplemented with 2.5 ng/ml basic FGF and supernatant of LIF-expressing cells) was added. The cells were cultured overnight before imaging. C2C12 cells were cultured in DMEM, 20% FBS, and HEK293 cells were cultured in DMEM, 10% FBS. For RNA interference, 2 µg siRNA was mixed with Lipofectamin RNAiMax (1:3) for 15 min, and the complex was added to the medium. For transfection of plasmids, 2 µg plasmid and Viafect reagent (Promega, Wisconsin, USA) were used.

For sphere culture, cells were dissociated by Trypsin-EDTA (Sigma-Aldrich, St. Louis, USA). The cells were plated on 35-mm dishes treated with anti-adherence rinsing solution (STEMCELL, Vancouver, Canada). After overnight culture, cells formed spontaneously clusters. Clusters were picked under a stereo microscope (Leica, Wetzlar, Germany) for imaging or differentiation analysis by flow cytometry (BD, New Jersey, United States). For chimeric sphere culture, transfected cells (*nGFP* and *EpDll1-NanoLuc* plasmids) and nontransfected cells were mixed at a ratio of 1:50. For analysis of differentiation of cells in spheres, the cell clusters were dissociated into single cells by Trypsin-EDTA and FACS analysis was performed afterwards to determine their Pax7 and MyoG expression. Briefly, after 10 min fixation at room temperature, the cells were incubated with guinea pig anti-Pax7, mouse anti-MyoG and chick anti-GFP overnight at 4 °C. Subsequently, cells were washed and secondary antibodies conjugated with Cy2, Cy3 or Cy5 were added for 1 h at room temperature. The cells were analyzed with a FACSAria II (BD, New Jersey, US) sorter. Wild-type cells not treated with primary antibodies were used as a negative control for gating. Results are presented as the percentage of MyoG+ and Pax7+ cells in the GFP+ cell fraction.

**Mouse strains**. The *Pax7IREScreERT2*, *Pax7nGFP*, *Dll1luc*, *Dll1type2*, *Dll1flox*, *Hes1flox*, and *MyoD* mutant mouse strains were used in our work[18,36,64–67]. Mice were maintained on a mixed 129/Sv and C57BL/6 genetic background. Routine genotyping was performed by PCR. To induce mutations using *Pax7IREScreERT2* in 9–13-week-old mice, 125 µg of tamoxifen per gram body weight (Sigma-Aldrich, St. Louis, USA) was injected every 24 h for five consecutive days. Cardiotoxin injuries were introduced by intramuscular injection of 40 µl Cardiotoxin (10 µM in PBS) into the TA. The contralateral TA injected with PBS served as an uninjured control. For proliferation analysis, EdU (50µg/g body weight) was injected i.p. into pregnant female or adult mice 3 h before analysis. All experiments were conducted according to the policies and regulations established by the Max-Delbrück-Center for Molecular Medicine (MDC), Germany, and the Mondor Institut of Biomedical Research (IMRB), France, and received ethical approval of the Regional Office for Health and Social Affairs Berlin and the Ethics Committee of the French Ministry.

**Chromatin immunoprecipitation PCR and qPCR**. Chromatin immunoprecipitation was performed using C2C12 cells, or HEK293 cells co-transfected with *phEF1α-Hes1* and *pCMV-MyoD* and enhancer constructs (*EF*, *EF−E*, and *EF−N*). Anti-MyoD or Anti-Hes1 antibodies were used for chromatin immunoprecipitation[68]. Cells were fixed and chromatin was sheared to ~200–1000 bp fragments by sonication. Protein A Dynabeads (ThermoFisher, Massachusetts, USA) were incubated with RIPA buffer (140 mM NaCl, 10 mM Tris pH8.0, 1 mM EDTA, 1% Triton-X100, 0.1% SDS, 1 mg/ml BSA) and MyoD or Hes1 antibodies overnight at 4ºC. As a control, pre-immune serum was used. After washing with RIPA buffer, 100 µg chromatin was added and incubated with antibody-coated beads overnight. Beads were washed and de-crosslinked by incubation at 65 ºC. DNA was purified and used for RT-PCR analysis. The sequences of primers used for the PCR reactions are listed in Supplementary Table 1. Consensus E- and N-boxes in the *Dll1* enhancer fragment *EF* were identified using CAGCTG (E-Box) and the CACCAG (N-box) as consensus sequences.

For quantitative PCR (qPCR), freshly isolated muscle stem cells were lysed, and total RNA was isolated using TRIzol reagent (Invitrogen, Massachusetts, USA). cDNA was synthetized using SuperScript III reverse transcriptase (ThermoFisher, Massachusetts, USA) after DNase treatment, and analyzed using SYBR qPCR Mix (ThermoFisher, Massachusetts, USA) and the CFX96 RT-PCR system (Bio-Rad, Hercules, USA). β-Actin was used as an internal standard.

**Plasmid construction and dual-luciferase reporter analysis**. To analyze the dynamics of Dll1 expression in spheres, *EpDll1-ub-NLS-NanoLuc* and *pCAG-nGFP* plasmids were generated by PCR and Gibson assembly. To analyze the enhancer activity of *EF*, *EF−E*, and *EF−N* fragments, luciferase was used as a reporter[69]. Briefly, the fragments were synthesized by Invitrogen (ThermoFisher, Massachusetts, USA) and inserted into *pGL4.23* (Promega, Wisconsin, USA). The *pRL-TK* plasmid (Promega, Wisconsin, USA) was used as a control for the transfection efficiency. Enhancer analysis was done according to the dual-luciferase reporter assay technical manual. HEK293 cells were co-transfected, lysed in 1× Passive Lysis Buffer (PLB) after 24 h of culture, and the lysates were transferred onto a white plate (Perkin Elmer, Massachusetts, USA). Luciferase activity was measured using a luminescence microplate reader (Berthold, Bad Wildbad, Germany).

**Organotypic slice culture**. E11.5 embryos were dissected in PBS and embedded in 4% low melt temperature agarose. 100 µm slices were prepared using a vibratome (Leica, Nussloch, Germany). Slices containing the limb were transferred to a glass-bottom dish containing 1 mg/ml collagen (Sigma-Aldrich, St. Louis, USA) in DMEM and neutralizing buffer for 10 min in a 37 °C incubator. Growth medium (DMEM, 10% Horse Serum, 0.5% Chicken Embryo Extract) with 1 µM luciferin (PJK GmbH, Kleinblittersdorf, Germany) was added, the dish was placed on the stage of an inverted microscope (IX83-ZDC, Olympus, Tokyo, Japan) and maintained at 37 °C in 5% CO$_2$. The nGFP signal from the *Pax7nGFP* transgene was used to focus and to track the cells. Bioluminescence was acquired using the EM-CCD camera (Hamamatsu, Shizuoka, Japan).

**Image analysis and quantification**. Image analysis was performed using Fiji software (ImageJ, v.2.1) and custom plug-ins[25]. The semi-automated tracking approach was used to determine cell locations and the intensity of the signal over time. Cells were identified using differences of Gaussian detection and the nearest neighbor searching approach in successive frames. This process can be corrected manually according to bright-field images. The mean intensity values of the signal of different single cells over time were recorded. After deduction of the background, 7th order Savitzky-Golay polynominal fitting was used to smooth the image. The oscillation periods were measured as the length of time between two peaks of bioluminescence signals. To define the stability of the oscillations, FFT with Hanning window was used by Origin software (OriginLab, Massachusetts, USA), which transfers a time-dependent function into its corresponding frequency-domain function. The dominant frequency can be identified as a peak. The powers of the frequency bands corresponding to periods between 1.5 and 3.75 h were quantified as area under the FFT curve[25]. Data from randomly chosen cells that could be followed over at least 10 h were analyzed. Matlab software (R2016a, MathWorks, Massachusetts, USA) was used to determine the phase relationship of

oscillations in two coupled cells. To derive accurate phase differences, we performed the following signal processing steps[51]: (i) detrending (by moving average with a window of 3 h); (ii) normalization (by dividing by a sliding standard deviation with a window of 3 h); (iii) Hillbert transformation analysis. The time at which the oscillatory signal in one cell corresponds to a peak was marked, and time intervals to the next peaks in oscillation of neighboring cells were determined.

**Mathematical modeling**. To model the network of Hes1, MyoD, and Dll1 in single cells, we extended our original qualitative ordinary differential equation (ODE) model of Hes1 and MyoD[25] to incorporate Dll1. In the original model, *Hes1* mRNA, Hes1 protein, a Hes1 interacting factor, *MyoD* mRNA, MyoD protein and MyoD interacting factor are included and it is assumed that Hes1 protein suppresses both *MyoD* and *Hes1* transcription. We extended this model, including *Dll1* mRNA and protein and the regulation of the *Dll1* transcription, i.e., the positive regulation by MyoD protein and the inhibition by Hes1 protein. For the investigation of the impact of changes of the *Dll1* transcription time, we established a delay differential equation model (DDE) for a single cell in which only Hes1 and Dll1 are included, which extends an earlier model of Shimojo et al.[36]. In this delay differential equation single-cell model, Hes1 inhibits synthesis of itself (with delay time $\tau_1$) as well as Dll1 production (with delay time $\tau_{21}$). This single-cell model was carefully mapped to the old model to preserve the dynamics of the variables. We then derived a coupled-cell model by combining and extending two single-cell models. In the coupled situation, Dll1 protein from one cell induces Hes1 synthesis in the second cell with a delay $\tau_{22}$. Details of all model structures and parameters are given in the Supplementary Methods.

**Statistical analysis**. Statistical analysis (Student's *t*-test) was performed using Excel software (Microsoft). *P* values < 0.05 were considered as significant and are shown in the figures.

**Reporting summary**. Further information on research design is available in the Nature Research Reporting Summary linked to this article.

## Data availability
Data are available in the Article, Supplementary Information or from the corresponding authors (C.B. and Y.Z.) upon reasonable request. Source data are provided with this paper.

## Code availability
The code used for the mathematical modeling is accessible from this GitLab repository: https://gitlab.com/kabaum/Mathematical_Model_of_Dll1_Hes1_MyoD.

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

## Acknowledgements
We are grateful to Petra Stallerow (MDC, Berlin) for help with animal husbandry. We also thank Hans-Peter Rahn for advice and help with cell sorting, Michael Strehle for providing the drawing shown in Fig. 9 and Elijah Lowenstein and Thomas Müller for critically reading the manuscript (all from the MDC, Berlin). This work was supported by grants of AFM Telethon, DFG (FOR 2841), ANR/DFG, and Helmholtz POF and AMPro to C.B., and by a Grant-in-Aid for Scientific Research on Innovative Areas from MEXT, Japan (16H06480) to R.K.

## Author contributions
Y.Z., R.K., and C.B. conceived the work and designed the project. Y.Z., I.L., and P.M. performed experiments. Y.Z and C.B. analyzed data. I.L. supplied technical guidance to Y.Z. K.B. and J.W. generated and analyzed the mathematical models. H. S. generated the Dll1^type2^ mouse mutant. C.B. with the support of all authors wrote the manuscript; all authors commented on the manuscript.

## Funding

## Competing interests
The authors declare no competing interests.
