## [Peer Review File · Nature Communications]

Reviewers' Comments:

Reviewer #1:

Remarks to the Author:

In this manuscript, Zhang et al. provide evidence that the oscillatory expression of the Notch signalling ligand Dll1 controls the muscle stem cell fate during developmental and regenerative myogenesis. This study follows up on their previous work exploring oscillatory Hes1 expression to control MyoD expression and activated stem cell expansion. Here the authors demonstrate that Hes1 binds to the enhancer of Dll1 as well, and represses its expression in an oscillatory manner allowing for the maintenance of self-renewal. However, loss of Hes1 expression impedes cyclic expression of MyoD and Dll1, leading to their sustained expression and, ultimately, muscle cell differentiation. Thus, dynamic expression of Hes1, Dll1 and MyoD regulates stem cell self-renewal and the timing of differentiation.

Although it was previously shown that Dll1 expression in differentiated cells provides self-renewing signal to the neighbouring cells, the real novelty of this paper lies in the fact that it is actually Dll1 oscillatory expression that allows for self-renewal maintenance. The experiments are well conducted. The conclusions are also strongly supported by the experiments that use different mouse models: to analyse Dll1 oscillation by live-imaging, to conditionally knockout Dll1 in muscle stem cells or to disrupt specifically its oscillation during myogenesis or adult muscle repair.

General comments

1. In the first figure (Fig 1A-F), the cell counting is presented in ratio while percentage might be more appropriate and easier to follow. Perhaps the authors could plot these data in a graph where the reader can see the progression in Dll1 expression at each of the 4 time points analysed during fibre culture. Finally, indicate the number of mice that have been analysed.
2. It is surprising that the regenerating myofiber at 7 dpi in the TxDll1f/f mice display similar diameter than controls while they contain less myonuclei (Fig 1K-O). Has the muscle phenotype been studied at later stages of the regeneration process (21 dpi for example)? One could expect to see a global decrease in muscle fibre size. As well, less myonuclei would suggest that less progenitors have fused to regenerate the myofibers. Have the authors tested the proliferative capacity of TxDll1f/f satellite cells using EdU assay for example? Is there any impairment?
3. At 7 dpi, has the number of myogenin+ cells been counted in the TxDll1f/f mice? It would be nice to have the same analysis in the TxDll1f/f mice than the one performed in the TxDll1f/type2 mice (Fig 5O-Q), further confirming that the oscillation of Dll1 is as important as its expression. The authors should provide the absolute number of both Pax7+ and myogenin+ cells per mm² as well as their proportions, at the same time point of regeneration.
4. Correlating with the latter comment, in Fig 1I and Fig 6K, providing the total number of cells per myofiber would help to confirm that Dll1 regulates the fate of activated stem cells rather than their expansion.
5. In Fig 2F, the authors should provide pictures the GFP channel to confirm that the analysed cells are GFP+ and expressing Pax7.
6. Please provide the citation justifying the choice of p/Mymk promoter as a positive control of the ChIP-qPCR in Fig 3C.
7. In Fig 6I, how was the number of satellite cells per myofiber calculated? The number of 20 Pax7+ cells per myofibers on cross-sections seem to be pretty high.

Reviewer #2:

Remarks to the Author:

The manuscript by Yao Zhang et al. identifies an important new oscillating component of Notch pathway in myogenic cells, the Notch ligand Dll1 and shows that Dll1 oscillations control the balance between self-renewal and differentiation of muscle stem cells. Authors analyze regulatory impact of Hes1 and MyoD (previously found to have oscillatory expression) on Dll1 oscillation in a community of myogenic cells. They identify Dll1 enhancer to which bind both Hes1 and MyoD and propose that Hes1 acts as transcriptional pacemaker regulating oscillations. In vivo analyses performed on individual but also on a

population of myogenic cells expressing Dll1-luc allowed to identify period of Dll1 oscillations and roles of Hes1 and MyoD.

Overall, this is a well-executed mechanistic study providing a new set of data and improving our understanding of oscillating Notch network in myogenic cells. Additional experiments or arguments are necessary to support some of author's conclusions.

Major points:

1. Authors claim (line 239): « Thus, MyoD and Hes1 enhance and repress Dll1 transcription, respectively, and directly bind to enhancer sequences in the Dll1 gene, yet they function independently of each other when controlling Dll1 expression. » Whether MyoD and Hes1 bind to distinct sequence motifs within Dll1 enhancer and thus act independently is not demonstrated. More information about distribution of potential E-boxes and N-boxes within 700bp of Dll1 enhancer and on extents of amplified fragments in ChIP-PCR experiments need to be provided. A mutagenesis of identified binding sites and co-ChIP experiments could help to clarify this issue.
2. The sentence in the abstract: « Dll1 oscillations are established by opposing functions of the Hes1 repressor and MyoD activator that bind the same Dll1 enhancer. » is misleading as MyoD appears dispensable for Dll1 oscillations which are still present in MyoD mutant context. With respect to MyoD function, are MyoD oscillations required for Dll1 regulation? Could sustained MyoD also positively regulate Dll1?
3. Authors provide evidence that oscillatory Dll1 produced by activated stem cells controls self-renewal of neighboring stem cells. This raises a question of whether Dll1 oscillations promote proliferative capacities of myogenic stem cells. Did authors test whether interfering with Dll1 oscillations affects muscle stem cells proliferation in developing and in regenerating muscle.
4. The observation that Dll1 and Dll1type2 both oscillate in newly activated Dll1type2 mutant muscle stem cells that are not in a stem cells community raises question about stimuli that could initiate oscillations but also about context or cell specific sensitivity of an oscillating network to a delay between transcription and translation. Could mathematical modeling provide some inputs?

Other points:

Fig. 1EF – one would expect to see a similar to Fig. 1A-D zoom allowing to appreciate co-expression of Dll1 transcripts with Pax7 and MyoD/MyoG in regeneration assay. Single channel need to be shown to demonstrate co-expression.

Not only initiation but also exit from oscillatory behavior appears important for biological outcomes. Could authors hypothesize how cell communities manage to move from oscillating to sustained expression of a gene network.

Reviewer #3:

Remarks to the Author:

The authors present a thorough analysis of the function of Dll1 oscillations in the developing and adult muscle. They propose a mechanism in which the balance between differentiation and self-renewal of neighbouring stem cells within a cell cluster is regulated by a combination of lateral inhibition and an induced oscillatory GRN. This is a highly relevant topic for researchers studying tissue homeostasis, but also embryonic development and regeneration. I recommend the manuscript by Zhang et al. for publication in Nature Communications, given the suggestions for revisions detailed below are addressed.

- One big conclusion is the role of Dll1 in interaction between neighbouring cells. To analyse this further, the authors should do the following:
 - o The authors have to quantify the phase relationship in neighbouring cells instead of just saying "anti-phase or out-of-phase" (l. 204). In addition, they should show data also as detrended + normalized to be able to visualize phase-relationship.
 - o When using the TxDll1f/type2 mice, in which the Dll1 oscillation period is increased by 6 min, single cells oscillate, while clusters of cells do not. The period of the Dll1 oscillations in single cells should be quantified. In addition, the effect on Hes1 oscillations in both single and clusters of cells should be determined. The authors should address what

implications these points have for the cell-cell communication?

o In Fig. S4 the authors quantify the variation of Hes1 levels in Pax7+ cells. In the analysis the authors should differentiate between single cells and clusters of stem cells.

o In immunostainings and stills of the movies a brightfield image should be displayed to show the number and organization of cells, especially because the point of the paper is to discuss the interaction between neighbouring stem cells.

o In imaging of wt cells, can the authors find examples in which one cell in a cluster permanently activates Dll1? The authors should determine how dynamics of cells in the same cluster look like and quantify.

o The authors should perform co-culture experiments of the different mutant lines they have to understand how oscillatory, randomly fluctuating or stable Dll1 expression affects dynamics and differentiation vs. proliferation in neighbouring cells. Such experiments could also be done by knockdown in one cell population and mixing this with control cells. Combined, these points will allow the authors to derive a clearer understanding on how the network of Dll1 and Hes1 controls proliferation and differentiation in cell populations. This should also be discussed in more detail at the end: What happens if one cell in a cluster differentiates (stable expression of Dll1 in one cell, continuous activation of Notch signalling in neighbouring cell, stable Hes1 oscillations...?)

- Whenever siRNA is performed, the authors confirm RNAi efficiency in C2C12 cells. RNAi efficiency should however be tested in the model system being analysed afterwards.

Especially since the knockdown is far from complete (already in C2C12 cells), immunostainings should be performed to test knockdown efficiency in single cells.

In case of Dll1 knockdown (lines 157 ff.) this does not affect the conclusions, since subsequently, a conditional knockout of Dll1 in Pax+ cells is performed.

Upon Hes1 knockdown in activated stem cells the authors find that Dll1 no longer oscillates stably. There are either random fluctuations or sustained expression. Does this correlate with the Hes1 knockdown efficiency? Myofibers should be stained for Hes1 expression after imaging to test this. Alternatively/ Additionally, the authors should speculate on why Dll1 levels still fluctuate. They should also address what the role of other bHLH proteins that are expressed in Pax7+ cells, such as Hes5 (see Fig. 5B), is?

Other things:

- The authors mention that "further work is required to identify the stimuli" that induce oscillations in single stem cells (lines 427-429). It would be useful to discuss this further. Are other Notch ligands known to be expressed etc.?

- Whenever Pax7+ cells are quantified, images of the Pax7 channel should be shown separately (otherwise it is difficult to match the images to their quantifications).

- The authors model the oscillatory network consisting of Hes1, MyoD and Dll1 as other delayed negative feedback systems have been modelled previously. While this makes sense, I cannot comment on the mathematical details of the model they derive.

- For the reader's convenience the figures should be restructured to e.g. spatially combine panels using the same model system (myofibers, isolated stem cells, mice etc.). In addition, links from the Methods section to the supplementary tables would be useful.

We thank all the reviewers for their constructive comments and appreciate their efforts to improve into our manuscript. To address the points raised, we performed a number of additional experiments. The new results helped us to strengthen our manuscript and support our previous findings. In particular, we added additional controls on the proliferation of myogenic cells in the Dll1 mutants, and provided additional evidence that the Dll1type2 mutation only affects oscillations of myogenic cells in communities but not in single cells. In summary, our results show that the mere presence of Dll1 does not suffice for the appropriate balance between self-renewal and differentiation of myogenic stem cells. Rather, oscillatory Dll1 input onto the Notch signaling cascade is needed for the correct balance. We indicate below our point-by-point responses to the comments of the reviewers in blue.

Point-by-point responses to the comments are provided below.

Reviewer #1 (Remarks to the Author):

In this manuscript, Zhang et al. provide evidence that the oscillatory expression of the Notch signalling ligand Dll1 controls the muscle stem cell fate during developmental and regenerative myogenesis. This study follows up on their previous work exploring oscillatory Hes1 expression to control MyoD expression and activated stem cell expansion. Here the authors demonstrate that Hes1 binds to the enhancer of Dll1 as well, and represses its expression in an oscillatory manner allowing for the maintenance of self-renewal. However, loss of Hes1 expression impedes cyclic expression of MyoD and Dll1, leading to their sustained expression and, ultimately, muscle cell differentiation. Thus, dynamic expression of Hes1, Dll1 and MyoD regulates stem cell self-renewal and the timing of differentiation.

Although it was previously shown that Dll1 expression in differentiated cells provides self-renewing signal to the neighbouring cells, the real novelty of this paper lies in the fact that it is actually Dll1 oscillatory expression that allows for self-renewal maintenance. The experiments are well conducted. The conclusions are also strongly supported by the experiments that use different mouse models: to analyse Dll1 oscillation by live-imaging, to conditionally knockout Dll1 in muscle stem cells or to disrupt specifically its oscillation during myogenesis or adult muscle repair.

We thank this reviewer for acknowledging the quality and depth of our work, and for their overall positive assessment. Indeed, the role of Dll1 was previously investigated by several researchers, whose work we cited in our manuscript. As the reviewer pointed out in their assessment, the novelty and impact of our work is not that Dll1 is functionally important but that *Dll1 oscillations and not its mere expression* controls self-renewal and differentiation of muscle stem cells in development and regeneration.

General comments

1. In the first figure (Fig 1A-F), the cell counting is presented in ratio while percentage might be more appropriate and easier to follow. Perhaps the authors could plot these data in a graph where the reader can see the progression in Dll1 expression at each of the 4

time points analysed during fibre culture. Finally, indicate the number of mice that have been analysed.

As the reviewer suggested, we present the cell counts as a graph displaying percentages, showing Dll1 expression in the distinct cell types (Supplementary Fig. 1c,f of the revised manuscript). The number of mice analyzed is now also indicated.

2. It is surprising that the regenerating myofiber at 7 dpi in the *TxDll1f/f* mice display similar diameter than controls while they contain less myonuclei (Fig 1K-O). Has the muscle phenotype been studied at later stages of the regeneration process (21 dpi for example)? One could expect to see a global decrease in muscle fibre size.

As suggested by the reviewer, we also analyzed a later stage of regeneration, and observed that the fiber diameter is significantly altered in *TxDll1f/f* and *TxDll1f/type2* mice at 21dpi. This is now shown in the Fig. 1g and Fig. 7e of the revised manuscript.

We agree with the reviewer that the fact that fiber diameters in the mutants were unchanged at 7 dpi was unexpected. To control for this, we repeated all quantifications at 7dpi using myosin antibodies (Figure 1 for the reviewers) to identify fibers. This excludes ghost fibers that might be counted when only a laminin staining is used. Quantification of the diameter using this method again showed that at 7dpi, the fiber diameters of the mutants and controls are not significantly different.

Figure 1 for reviewers: Histological analysis and quantification of the fiber diameter using pan-Myosin and Collagen IV antibodies. DAPI was used as counterstain to identify newly generated fibers with centrally located nuclei. WE analyzed control and mutant muscles at an early stage of regeneration (7dpi).

As well, less myonuclei would suggest that less progenitors have fused to regenerate the myofibers. Have the authors tested the proliferative capacity of *TxDll1f/f* satellite cells using EdU assay for example? Is there any impairment?

This reviewer and reviewer #2 (see below) suggested to analyze proliferation of mutant muscle stem cells, which we performed in the revised version. The results show no difference in the rate of EdU incorporation into Pax7+ cells at early stages of regeneration (Supplementary Fig. 1j, 7b of the revised manuscript) or in the developing muscle at E14.5 (Fig. 8a of the revised manuscript). To make the quantification easier, we used 3dpi and not 4dpi to analyze proliferation during regeneration. Due to the uncontrolled differentiation, Pax7+ cells are rare at 4dpi.

Our preliminary data indicated that there are less than 2 EdU+/Pax7+ cells/mm² at this stage. At 3 dpi, we observed considerably more Pax7+ cells, which aided quantifications.

3. At 7 dpi, has the number of myogenin+ cells been counted in the TxDll1f/f mice? It would be nice to have the same analysis in the TxDll1f/f mice than the one performed in the TxDll1f/type2 mice (Fig 5O-Q), further confirming that the oscillation of Dll1 is as important as its expression. The authors should provide the absolute number of both Pax7+ and myogenin+ cells per mm² as well as their proportions, at the same time point of regeneration.

We thank the reviewer for pointing this out to us. Indeed, we had displayed the data in an unfortunate and non-equivalent manner in the original manuscript. We do now provide equivalent datasets, i.e. an analysis at the same stages of regeneration, and the use of absolute numbers of both, Pax7+ and MyoG+ cells/mm² (Fig. 1f, 7d of the revised manuscript). This allows for a direct comparison of TxDll1f/f and TxDll1f/type2 phenotypes, substantiating our claim that not only the presence of Dll1 but also its dynamic expression is functionally important.

4. Correlating with the latter comment, in Fig 1I and Fig 6K, providing the total number of cells per myofiber would help to confirm that Dll1 regulates the fate of activated stem cells rather than their expansion.

To address this concern, we counted numbers of cells in the colonies of control and Dll1 mutants. We observe the same phenomenon as in vivo, i.e. no effect on colony size before the premature differentiation sets in. Thus, on the fiber at early stages of culture (60 h), the colony size is identical no matter whether the fibers were isolated from control or Dll1 mutants. However, at 72 h muscle stem cells in the colonies of Dll1 mutants have differentiated prematurely, as evident from the increased proportion of MyoG+ cells. At this time of culture, we observed a change in the size of the colonies. We display these additional data in Fig. 1c and 7a of the revised manuscript.

5. In Fig 2F, the authors should provide pictures the GFP channel to confirm that the analysed cells are GFP+ and expressing Pax7.

As requested by the reviewer, we now provide a picture of the GFP channel to show that the cells analyzed are Pax7+ (revised supplemental Fig. 2f). The reviewer should also note that all Dll1+ cells are myogenic in the limb at E11.5, i.e. the developmental stage analyzed here. Endothelial cells also express Dll1, but only turn the gene on at later stages (our own observations and Fig.1 in a previous report (Sorensen et al., 2009)).

6. Please provide the citation justifying the choice of p/Mymk promoter as a positive control of the ChIP-qPCR in Fig 3C.

We include the appropriate citation in the revised manuscript (Millay et al., 2014).

7. In Fig 6I, how was the number of satellite cells per myofiber calculated? The number of 20 Pax7+ cells per myofibers on cross-sections seem to be pretty high.

We thank the reviewer for pointing out this mistake; the number of Pax7+ cells on the sections was determined per 100 fibers, but the labeling was wrongly given as cell numbers/per fiber. The mistake has been corrected in the revised version of our manuscript (Fig. 8b of the revised manuscript).

Reviewer #2 (Remarks to the Author):

The manuscript by Yao Zhang et al. identifies an important new oscillating component of Notch pathway in myogenic cells, the Notch ligand Dll1 and shows that Dll1 oscillations control the balance between self-renewal and differentiation of muscle stem cells. Authors analyze regulatory impact of Hes1 and MyoD (previously found to have oscillatory expression) on Dll1 oscillation in a community of myogenic cells. They identify Dll1 enhancer to which bind both Hes1 and MyoD and propose that Hes1 acts as transcriptional pacemaker regulating oscillations. In vivo analyses performed on individual but also on a population of myogenic cells expressing Dll1-luc allowed to identify period of Dll1 oscillations and roles of Hes1 and MyoD.

Overall, this is a well-executed mechanistic study providing a new set of data and improving our understanding of oscillating Notch network in myogenic cells. Additional experiments or arguments are necessary to support some of author's conclusions.

We thank the reviewer for carefully reading our manuscript and judging our work as 'well-executed'. We appreciate the overall positive assessment.

Major points:

1. Authors claim (line 239): « Thus, MyoD and Hes1 enhance and repress Dll1 transcription, respectively, and directly bind to enhancer sequences in the Dll1 gene, yet they function independently of each other when controlling Dll1 expression. » Whether MyoD and Hes1 bind to distinct sequence motifs within Dll1 enhancer and thus act independently is not demonstrated. More information about distribution of potential E-boxes and N-boxes within 700bp of Dll1 enhancer and on extents of amplified fragments in ChIP-PCR experiments need to be provided. A mutagenesis of identified binding sites and co-ChIP experiments could help to clarify this issue.

As suggested by the reviewer, we now analyzed the MyoD/Hes1 binding sites in the Dll1 intron, i.e. the regulatory sequences that possesses enhancer activity. In the original submission, we had employed a large fragment (>700bp) for the analysis of the enhancer activity, and had only amplified a part of it (151 bp fragment) in ChIP-PCR experiments in C2C12 cells. In the revised manuscript, we now consistently use this 151 bp fragment (EF) and two synthesized mutant fragments, one lacking all MyoD binding sites (EF^{-E}; CAGCTG replaced by CAGtTt), and a second lacking the Hes1 binding site (EF^{-N}; CACCAG replaced by CAaaAG). The wildtype EF functions as a regulatory sequence and responds to MyoD, MyoG and Hes1: luciferase expression controlled by EF is regulated positively by MyoD (8.9 fold), MyoG (3.0 fold) and negatively by Hes1 (0.6 fold). The EF^{-E} sequence no longer enhances transcription in the presence of MyoD, but still responds to Hes1. Conversely, the

EF^N sequence no longer responds to Hes1, but MyoD still activates transcription driven by this fragment. The new data are now described on pg. 10 and shown in Fig. 3g of the revised manuscript. In addition, we provide ChIP-PCR data to demonstrate that MyoD and Hes1 bind independently of each other, i.e. the wildtype fragment binds both, MyoD and Hes1, whereas the EF^N sequence binds MyoD but not Hes1, whereas the EF^E sequence binds Hes1 but not MyoD (Supplementary Fig. 3c of the revised manuscript).

2. The sentence in the abstract: « Dll1 oscillations are established by opposing functions of the Hes1 repressor and MyoD activator that bind the same Dll1 enhancer. » is misleading as MyoD appears dispensable for Dll1 oscillations which are still present in MyoD mutant context. With respect to MyoD function, are MyoD oscillations required for Dll1 regulation? Could sustained MyoD also positively regulate Dll1?

This was an ambiguous sentence, and we corrected it. Indeed, all our data as well as the mathematical model show that MyoD only controls the expression levels, but its oscillations are dispensable for Dll1 oscillations. We have rewritten the abstract to express this in an unambiguous manner. The new sentences of the abstract reads: *We show that Dll1 oscillations are controlled through a Dll1 enhancer that is bound by the Notch target Hes1 and the muscle regulatory factor MyoD. Consistent with our mathematical model, Hes1 acts as the oscillatory pacemaker in the network, whereas MyoD regulates robust Dll1 expression.*

3. Authors provide evidence that oscillatory Dll1 produced by activated stem cells controls self-renewal of neighboring stem cells. This raises a question of whether Dll1 oscillations promote proliferative capacities of myogenic stem cells. Did authors test whether interfering with Dll1 oscillations affects muscle stem cells proliferation in developing and in regenerating muscle.

This comment is related to point #2 of reviewer #1. We performed EdU-labeling experiments to quantify proliferation. The results show no difference in the rate of EdU incorporation into Pax7+ cells in the regenerating (Supplementary Fig. 1j,7b of the revised manuscript) or in the developing muscle (Fig. 8a of the revised manuscript). In addition, we determined colony sizes in floating fiber cultures. We observe no effect on colony size before premature differentiation sets in. Thus, on fibers at early stages of culture (60 h), the colony size is identical no matter whether the fibers were isolated from control or Dll1 mutants. However at 72 h, muscle stem cells in the colonies of Dll1 mutants have differentiate prematurely. This is evident from the increased proportion of MyoG+ cells and the decrease in the colony size. We display these additional data in Fig. 1c,7a of the revised manuscript.

4. The observation that Dll1 and Dll1type2 both oscillate in newly activated Dll1type2 mutant muscle stem cells that are not in a stem cells community raises question about stimuli that could initiate oscillations but also about context or cell specific sensitivity of an oscillating network to a delay between transcription and translation. Could mathematical modeling provide some inputs?

In order to further model the influence of the change in the time required for transcription (i.e. the change introduced by the Dll1type2 mutation), and to predict the behavior of uncoupled versus coupled cells, we extended our modeling

framework, deriving an additional delay differential equation model of the system. This allows us to assess the impact of the transcriptional delay of Dll1type2 in single cells as well as in two coupled cells. The reviewer should note that a number of theoretical papers already show that a two-cell model can accurately capture the characteristics of networks of higher numbers of coupled cells (Yoshioka-Kobayashi et al., 2020). We describe the model in detail in the Supplementary Methods section.

Single Cell Models: Dll1 vs Dll1type2

We originally modeled the regulatory network of all three components, MyoD, Hes1 and Dll1 by an ordinary differential equation model which was already shown in the original manuscript. During the revision, we additionally developed a delay differential model (subsequently called our new model), building on the framework of a previous model reported by Shimojo et al. (Shimojo et al., 2016) which was extended by introducing Dll1. We mapped the dynamics of the new single cell model and compared it to the ordinary differential equation model previously used (Fig. 3 of the originally submitted manuscript. The comparison of the dynamics of the 2 models is shown in Fig.1 in Supplementary Methods). The two models give similar results and do thus equally well describe the gene expression dynamics.

We used the new model to assess in a single cell the effect of an increased time needed for transcription in the Dll1type2 mutant (i.e. 0.1h or 6 minutes) and therefore also the time needed for the production of the encoded protein. The 0.1h change had been experimentally determined by Shimojo et al (Shimojo et al., 2016). They used an optogenetic promoter to induce wildtype and Dll1type2 genes in cultured cells. The increase is predicted to cause a change in the phase shift between Hes1 and Dll1 (time difference between maxima of Dll1 and Hes1 expression) from 1.9h (wildtype) to 2.0h (mutant, Dll1type2). However, the oscillatory stability, oscillatory period or amplitudes were predicted to be unaffected (Fig S5d). The text describing the model was introduced on pg. 12 and 13 of the revised manuscript.

Coupled cell models: Dll1 vs Dll1type2

We then extended the newly derived single cell model to include cell coupling and to predict the behavior of two coupled cells (shown schematically in Fig. 5c). The delays τ_1 and τ_{21} are identical to those used in the single cell model. We estimated the inter-cellular delay (τ_{22}), i.e. the time between appearance of Dll1 protein in one cell and the appearance of Hes1 protein in the other cell, to be on the order of 1.3 hr. This value is based on experimental measurements reported previously (Isomura et al., 2017). For two coupled wildtype cells, the model predicts out-of-phase oscillations (shown on the left of Fig. 5d). In the Dll1type 2 mutant, the oscillatory amplitude is very severely quenched (shown in Fig. 5d, middle panel). In summary, our new model is in full accordance with the results of our experiments. In particular, it supports the notion that the change time needed for Dll1type2 transcription can severely affect the entire oscillatory system in coupled cells.

Because of a comment of reviewer #3 (who suggested co-culture experiments using the different lines) we also modeled one wildtype and one Dll1type2 mutant cell that are coupled to each other. Stable oscillations were predicted for such 'chimeric situations', albeit with a lower oscillatory amplitude than the one predicted for two

coupled wildtype cells (Fig. 5d, right panel). The reviewer should note that in the revised manuscript we experimentally assessed such a chimeric situation in mixed sphere cultures using a transfected NanoLuc indicator that is driven by the Dll1 promoter/enhancer (EpDll1-NanoLuc) to visualize dynamic expression. During transfection different cells take up variable copy numbers of the plasmids, therefore we cannot use this approach to determine the amplitude of oscillations (see also response to reviewer 3).

The predictions of the single cell model about entry/exit into quiescence or entry into differentiation are discussed below.

Other points:

Fig. 1EF – one would expect to see a similar to Fig. 1A-D zoom allowing to appreciate co-expression of Dll1 transcripts with Pax7 and MyoD/MyoG in regeneration assay. Single channel need to be shown to demonstrate co-expression.

In the revised manuscript, we adjusted the zoom of Fig. 1b (Fig.1EF in original MS) to be similar to the zoom of the revised Fig. 1a (Fig.1A-D in original MS). We also provide the separated channel figures and a brightfield picture in the revised supplementary Fig. 1e.

Not only initiation but also exit from oscillatory behavior appears important for biological outcomes. Could authors hypothesize how cell communities manage to move from oscillating to sustained expression of a gene network.

Single Cell Model and Quiescence/Differentiation

Using the single cell ordinary differential equation model, we can assess which variables would need to change in order to allow entry into quiescence (i.e. a situation characterized by high Hes1, low or no Dll1 and MyoD) or entry into differentiation (low Hes1, high MyoD/Dll1).

[Redacted]

[Redacted]

The model indicates that changes that result in increased Hes1 protein levels, i.e. increased Hes1 transcription or protein synthesis rates as well as decreased Hes1 mRNA degradation rate result in a situation that resembles a 'quiescent' state, i.e. high sustained Hes1, and low MyoD and Dll1. Conversely, changes that result in low Hes1 protein levels, i.e. decreased Hes1 transcription or protein synthesis rates as well as increased Hes1 mRNA degradation rate results in a situation that resembles entry into differentiation. *[Redacted]* In conclusion, many options exist, and further work is needed to determine which of these parameters

change during entry into quiescence or terminal differentiation, and if such changes would suffice to account for the different cellular behaviors.

Reviewer #3 (Remarks to the Author):

The authors present a thorough analysis of the function of Dll1 oscillations in the developing and adult muscle. They propose a mechanism in which the balance between differentiation and self-renewal of neighbouring stem cells within a cell cluster is regulated by a combination of lateral inhibition and an induced oscillatory GRN. This is a highly relevant topic for researchers studying tissue homeostasis, but also embryonic development and regeneration. I recommend the manuscript by Zhang et al. for publication in Nature Communications, given the suggestions for revisions detailed below are addressed.

We thank the reviewer for the overall positive assessment of our work, and for pointing out that the mechanism we are analyzing is a ‘highly relevant topic for researchers studying tissue homeostasis, but also embryonic development and regeneration’.

- One big conclusion is the role of Dll1 in interaction between neighbouring cells. To analyse this further, the authors should do the following:
o The authors have to quantify the phase relationship in neighbouring cells instead of just saying “anti-phase or out-of-phase” (l. 204). In addition, they should show data also as detrended + normalized to be able to visualize phase-relationship.

As suggested by the reviewer, we detrended and normalized the luciferase tracks to define the phase relationship. For phase analysis, the following procedure was performed: 1. Detrending with a moving average (window of 3h). 2. Normalization with sliding standard deviation (window of 3h). Afterwards, Hilbert transformation was performed to plot the phase difference.

This analysis shows that on average the oscillations in two coupled cells are out of phase, and the average phase shift is around half of an oscillatory period. The data are displayed in the revised supplementary Fig. 2c.

o When using the TxDll1f/type2 mice, in which the Dll1 oscillation period is increased by 6 min, single cells oscillate, while clusters of cells do not. The period of the Dll1 oscillations in single cells should be quantified.

This is a misunderstanding: The Dll1type2 mutation prolongs the transcription time of Dll1 by 0.1 hours (6 minutes), and therefore affects the time that Hes1 requires to affect Dll1 protein levels. The mutation does not affect the oscillatory period (Fig. 6b). In the revised version, we considerably extended the modeling approach in order to show how the Dll1type2 mutation affects delays and to predict how this affects the oscillatory network. The revised Fig. 5, supplementary Fig. 5 and new text on pages 12-13 are now exclusively devoted to explain the models, to introduce the change in transcription and its effect on the delay, and to show the prediction on the oscillatory

behavior. Furthermore, we show the experimental verification of the predictions in the revised Fig. 6 of the manuscript.

In short, the Dll1type2 mutation prolongs the transcription time of Dll1 by 0.1h, and therefore prolongs τ_{21} , the delay time needed for Hes1 to affect Dll1 protein levels (see scheme in the revised Fig. 5c). This change in time had been experimentally determined by Shimojo et al (Shimojo et al., 2016) using an optogenetic promoter to induce Dll1luc and Dll1type2 gene constructs in cultured cells, and measuring the appearance of the protein. The model predicts that the oscillatory period is unaffected by the mutation in single cells. We verified this experimentally, and found no significant difference between the oscillatory period in single control and Dll1type2 mutant cells (revised Fig. 6c). Furthermore, quantification of the oscillatory stability (i.e. power determined from FFT) showed no significant difference (revised Fig. 6c). In contrast, the oscillatory amplitude in coupled Dll1type2 cells is predicted by the model to be severely quenched, and experimentally we do not observe stable oscillations (modelled in the revised Fig. 5d and experimentally verified in revised Fig. 6c,d).

In addition, the effect on Hes1 oscillations in both single and clusters of cells should be determined. The authors should address what implications these points have for the cell-cell communication?

To address this point of the reviewer, we analyzed the variance of Hes1 protein by distinguishing single and coupled myogenic cells in the regenerating muscle. Thus, Pax7+ cells that were or were not directly contacted by other myogenic cells were identified, and the Hes1 expression levels were compared. This demonstrated that Hes1 variance was similar in single Pax7+ cells from control and TxDll1^{f/type2} mice, but distinct in coupled Pax7+ cells, supporting the notion that the effects of the mutation depend on the coupling of the cells (Fig. 7d of the revised manuscript).

A more direct analysis of the effects of the Dll1type2 mutations on Hes1 oscillations would be an interesting experiment to do, but it is not easily feasible. When we began to analyze oscillations in the myogenic system we made some unsuccessful attempts to use fluorescent dyes to monitor protein expression. For instance, we had obtained a Hes1-Venus indicator line from the Kageyama group, but due to high levels of autofluorescence in the muscle we were unable to use it for imaging of muscle stem cells associated with fibers. Therefore, we used luciferase fusions in all constructs. The Dll1type2 allele (shown in Fig. 6a of the revised manuscript) contains Dll1-luc cDNA inserted into the first exon of Dll1 gene. Thus, the time for transcription of the primary transcript is increased, but the allele simultaneously allows for the monitoring of Dll1 expression dynamics. This also means that luciferase is already expressed in Dll1type2 mutant cells, i.e. we cannot use the Hes1-luciferase allele to directly image Hes1 oscillations.

o In Fig. S4 the authors quantify the variation of Hes1 levels in Pax7+ cells. In the analysis the authors should differentiate between single cells and clusters of stem cells.

As suggested by the reviewer, we quantified the variation of Hes1 levels in Pax7+ cells in single and clustered cells in the regenerating muscle in vivo (revised Fig. 7d). The results further supported the notion that the TxDll1type2 mutation does only

affects clustered cells. In single cells, the variation of Hes1 in Pax7+ cells was not significantly different in the regenerating muscle of TxCon and TxDll1f/type2 mutant mice ($P=0.64$). However, in coupled cells the variation of Hes1 was smaller in TxDll1type2 mutants ($P=0.02$).

o In immunostainings and stills of the movies a brightfield image should be displayed to show the number and organization of cells, especially because the point of the paper is to discuss the interaction between neighbouring stem cells.

As requested by the reviewer we now included brightfield images in the immunostainings and brightfield pictures of the cells that were tracked in the movies in the revised manuscript. These are either included in the main or the Supplementary Figures (Fig. 1a and supplementary Fig. S1d,1e for Dll1 expression analysis; Fig. 2c and supplementary Fig. S2a, 2b for tracks of Dll1 in control cells; supplementary Fig. 4a,b,c,d,e for tracks of imaged cells lacking MyoD and/or Hes1; supplementary Fig. S6b,6c,6d for the tracks in Dll1type2 mutant cells).

o In imaging of wt cells, can the authors find examples in which one cell in a cluster permanently activates Dll1? The authors should determine how dynamics of cells in the same cluster look like and quantify.

For this analysis, we used stem cells on floating fibers, because it models well self-renewal and differentiation reflecting an in vivo situation. In such cultures, the stem cells behave in a stereotypic manner. They are present as single activated cells after 12 h culture, divide once around 40 h, are then undifferentiated and co-express Pax7 and MyoD. We do not observe sustained Dll1 when we image activated single cells (after 24 h culture) or coupled two-cell colonies (after 48 h culture). We do now introduce the fiber cultures and the stereotypic behavior of the cells on the fibers on page 6 of the revised manuscript. After 72 h hours culture, the colony size and differentiation of the cells is heterogenous. Some are expected to display sustained Dll1 (e.g. MyoG+ cells), but we cannot image single cells in such clusters. The cells are motile and constantly change their relative positions in the cluster; since the luciferase exposure time is long we cannot follow them reliably (mentioned on pg. 8 of the revised manuscript).

o The authors should perform co-culture experiments of the different mutant lines they have to understand how oscillatory, randomly fluctuating or stable Dll1 expression affects dynamics and differentiation vs. proliferation in neighbouring cells. Such experiments could also be done by knockdown in one cell population and mixing this with control cells.

This is an interesting point that we addressed by mathematical modeling and experimentally by the use of sphere cultures. The model predicted very strong quenching of the oscillation in coupled Dll1type2 mutant compared to wildtype cells. In contrast, moderate quenching of oscillations was predicted for chimeric situations, i.e. coupled cells in which one cell is wildtype and the second cell is Dll1type2 mutant (the modeling results of this are shown in the revised Fig. 5d).

We introduced a novel sphere culture system to address this point experimentally. In sphere cultures, cells are in constant contact despite the fact that they change relative positions to each other. This is not the case in adherent cultures, where the

highly motile muscle stem cells have to be kept in non-confluent conditions to keep them healthy. To monitor Dll1 oscillations in the sphere, cells were co-transfected with plasmids containing a Dll1 promoter driving a cDNA that encodes a destabilized NanoLuc (NanoLuc uses a distinct substrate from firefly luciferase and can therefore be monitored independently of the luciferase activity of the Dll1type2 allele) and a CAG promoter driving EGFP to identify the transfected cells. We mixed transfected and untransfected cells at a ratio of 1:50, and monitored NanoLuc expression in EGFP+ cells in spheres containing (i) a transfected wildtype cell surrounded by wildtype cells and (ii) a transfected Dll1type2 mutant cell surrounded by Dll1type2 mutant cells. We observed the same oscillatory behavior as the one in coupled cells on fibers, i.e. stable oscillations in control and small fluctuations but no stable oscillations in Dll1type2 cells (Fig. 2c and 6d of the revised manuscript). This substantiates our results obtained in fiber culture, and extended our analysis of Dll1 expression dynamics to clusters containing more than two cells. The text describing the sphere culture experiments was introduced on pg. 9.

As the reviewer suggested, we used this for the analysis of mixed (chimeric) spheres containing (iii) a transfected wildtype cell surrounded by Dll1type2 mutant cells or (iv) a transfected Dll1type2 mutant cell surrounded by wildtype cells. In both situations, the transfected cells display stable oscillations (Supplementary Fig. 6 of the revised manuscript) which is also predicted by the mathematical model. The text describing the chimeric sphere culture experiments was introduced on pg. 13 and 14.

The reviewer also suggests addressing functions of random fluctuations versus stable expression. Random fluctuations and/or stable expression was the term that we had used in the original submission to describe Dll1 expression dynamics in situations when stable oscillations were not observed (i.e. Hes1 siRNA and Dll1type2 mutation). In the revised manuscript, we now show that such patterns are also observed in Hes1 null-mutant cells (Fig. 4b of the revised manuscript). Thus, the pattern does not correlate with the efficacy of an Hes1 siRNA knockdown. Random fluctuations have small amplitudes and no periodicity. For instance, we observe in one cell for instance a period of small fluctuations, followed by a period of sustained expression, then again small fluctuations. We consider this 'noise'. Noise in mRNA/protein level of genetically identical cells is well described in bacteria, yeast and mammalian cells (Paulsson, 2004; Raser and O'Shea, 2005). To assess the response to randomly fluctuating or stable Dll1 is thus impossible since we have not observed cells that expresses either randomly fluctuating or stable Dll1, rather these two states seem to be interchangeable. In the revised manuscript we explicitly state this on page 11. Further, we describe this dynamic pattern now as sustained expression interrupted by small fluctuations (pg. 11,13 and 14 of the revised manuscript).

Combined, these points will allow the authors to derive a clearer understanding on how the network of Dll1 and Hes1 controls proliferation and differentiation in cell populations. This should also be discussed in more detail at the end: What happens if one cell in a cluster differentiates (stable expression of Dll1 in one cell, continuous activation of Notch signalling in neighbouring cell, stable Hes1 oscillations...?)

- Whenever siRNA is performed, the authors confirm RNAi efficiency in C2C12 cells. RNAi efficiency should however be tested in the model system being analysed afterwards.

Especially since the knockdown is far from complete (already in C2C12 cells), immunostainings should be performed to test knockdown efficiency in single cells. In case of Dll1 knockdown (lines 157 ff.) this does not affect the conclusions, since subsequently, a conditional knockout of Dll1 in Pax+ cells is performed.

To address the concerns of the reviewer about Hes1 siRNA efficacy in stem cells on myofibers, we performed two independent approaches: i) we used a Hes1 null mutation, and performed quantifications of effects of the null mutation and Hes1 siRNA. Both equally interfere with stable Dll1 oscillations (revised Fig. 4b for Hes1 null mutation). Breeding of Hes1/MyoD double mutants could not be achieved during the revision time, and therefore Hes1 siRNA was used to knockdown Hes1 in MyoD^{-/-} cells in the revised manuscript (revised Fig. 4d and page 11 of result section). ii) In addition, we investigated Hes1 level in Pax7⁺ muscle stem cells on myofibers treated with control and Hes1 siRNA. Compared to control siRNA treatment, Hes1 protein levels after Hes1 siRNA treatment was 12.9%±1.1% (Supplementary Fig. 3a). For technical reasons we were unable to determine Hes1 levels directly in those cells that were imaged after siRNA treatment. We float myofibers and associated stem cells during incubation and imaging to prevent migration of the stem cells onto the culture plate. More than one fiber is present in the imaging chamber, and it is not possible to identify which one we imaged since they move when the chamber is removed from the microscope.

To address the concern about the Dll1 siRNA experiments, we removed them and only use conditional Dll1 null mutations in the revised manuscript .

Upon Hes1 knockdown in activated stem cells the authors find that Dll1 no longer oscillates stably. There are either random fluctuations or sustained expression. Does this correlate with the Hes1 knockdown efficiency? Myofibers should be stained for Hes1 expression after imaging to test this. Alternatively/ Additionally, the authors should speculate on why Dll1 levels still fluctuate.

The effects of a Hes1 null mutation and Hes1 siRNA on Dll1 oscillations are identical. Both equally interfere with stable Dll1 oscillations (Fig. 4d and page 11 of the revised manuscript). Thus, random fluctuation or sustained expression are not caused by different extents of the Hes1 siRNA knockdown. Random fluctuations have small amplitudes and no periodicity and seem random. For instance, in a single cell we observed small fluctuations, followed by a period of stable expression, then again small fluctuations. Thus, randomly fluctuating or stable Dll1 are not alternative stable states, and we consider these patterns as 'noise'. Noise in mRNA/protein level of genetically identical cells is well described in bacteria, yeast and mammalian cells (Paulsson, 2004; Raser and O'Shea, 2005). We discuss this in the revised manuscript on page 11.

They should also address what the role of other bHLH proteins that are expressed in Pax7+ cells, such as Hes5 (see Fig. 5B), is?

In general, Hes1 phenotypes in the muscle are weaker than those observed after RBPj mutation, raising the possibility that other Hes/Hey factors may function redundantly with Hes1. Such redundant functions have been described in

neurogenesis where Hes1, Hes3 and Hes5 cooperate (Imayoshi et al., 2010). We compared the role of a number of the genes (Hes1, Heyl, Hes5, Hes7) that are induced by Notch signaling in myogenic cells in our previous study (Lahmann et al., 2019). Among these, Hes1 mutants show the strongest phenotype, whereas the other mutants display none or very subtle phenotypes, and we therefore refrained from further genetic analysis. To our knowledge, nobody has assessed whether Heyl, Hes5, Hes7 also oscillate in muscle stem cells. So, we simply do not know whether other Hes/Hey factors act redundantly and, if they act redundantly, whether and how they contribute to the control of dynamic gene expression in myogenesis. We have introduced a sentence that refers to the previous comparison of Hes/Hey factors (supplemental Fig. 1, Lahmann et al. 2019) on page 13 of the revised manuscript.

Other things:

- The authors mention that "further work is required to identify the stimuli" that induce oscillations in single stem cells (lines 427-429). It would be useful to discuss this further. Are other Notch ligands known to be expressed etc.?

In the discussion on page 17, we included additional thoughts on how entry/exit into quiescence might be controlled by Notch signaling. Dll4, Jag1 and Jag2 are expressed by the myofibers that contact muscle stem cells. Recent work implicate increased Mib1 (Mindbomb 1) expression in the myofiber as one mechanisms that enhances Dll4 signaling during entry into quiescence (Kim et al., 2016). How exit from quiescence might be regulated is less clear, but changes in Hes1 stability/transcription rates/levels might well play a role there (see also comment of reviewer 2 and Fig. 2 for the reviewer)

- Whenever Pax7+ cells are quantified, images of the Pax7 channel should be shown separately (otherwise it is difficult to match the images to their quantifications).

In order to address this point, we introduced the following changes: (i) we changed the assigned colors in Fig. 1g, 7e and supplementary Fig. S1i), using green for Pax7, red for DAPI and blue for Collagen IV; (ii) we split the panels, displaying in the upper part Pax7/DAPI/ Collagen IV and in the lower panel Pax7 only. We hope that this satisfies the concern of the reviewer.

- The authors model the oscillatory network consisting of Hes1, MyoD and Dll1 as other delayed negative feedback systems have been modelled previously. While this makes sense, I cannot comment on the mathematical details of the model they derive.

Mathematical modeling of the network extends previous published work of the Kageyama and Birchmeier laboratories (Lahmann et al., 2019; Shimojo et al., 2016). Further, the parameters used in the model rely on previous published data. This is described in detail in the Supplementary Materials modeling.

- For the reader's convenience the figures should be restructured to e.g. spatially combine panels using the same model system (myofibers, isolated stem cells, mice etc.). In addition, links from the Methods section to the supplementary tables would be useful.

To address the concern of the reviewer, we restructured some of the paper and changed the text to improve readability: The general outline is (i) Dll1 expression and function in myogenic cells (Fig. 1), (ii) Dll1 oscillates (Fig. 2), (iii) experimental characterization of the regulatory network, (Figs. 3 and 4), (iv) mathematical models of Dll1 oscillations in single/coupled wildtype and Dll1type2 mutant cells (Fig. 5), (v) experimental verifications of changes in Dll1type2 mutant cells (Fig. 6) (vi) phenotypes of Dll1type2 mutants (Figs. 7 and 8). We discussed the option to spatially combine model system (myofibers, isolated stem cells, mice etc.) in v and vi but felt that this would become very redundant. I hope that the reviewer will find the new version more convenient.

References:

- Hirata, H., Yoshiura, S., Ohtsuka, T., Bessho, Y., Harada, T., Yoshikawa, K., and Kageyama, R. (2002). Oscillatory expression of the bHLH factor Hes1 regulated by a negative feedback loop. *Science* 298, 840-843.
- Imayoshi, I., Isomura, A., Harima, Y., Kawaguchi, K., Kori, H., Miyachi, H., Fujiwara, T., Ishidate, F., and Kageyama, R. (2013). Oscillatory control of factors determining multipotency and fate in mouse neural progenitors. *Science* 342, 1203-1208.
- Imayoshi, I., Sakamoto, M., Yamaguchi, M., Mori, K., and Kageyama, R. (2010). Essential Roles of Notch Signaling in Maintenance of Neural Stem Cells in Developing and Adult Brains. *J Neurosci* 30, 3489-3498.
- Isomura, A., Ogushi, F., Kori, H., and Kageyama, R. (2017). Optogenetic perturbation and bioluminescence imaging to analyze cell-to-cell transfer of oscillatory information. *Gene Dev* 31, 524-535.
- Kim, J.H., Han, G.C., Seo, J.Y., Park, I., Park, W., Jeong, H.W., Lee, S.H., Bae, S.H., Seong, J., Yum, M.K., *et al.* (2016). Sex hormones establish a reserve pool of adult muscle stem cells (vol 18, pg 930, 2016). *Nat Cell Biol* 18, 1109-1109.
- Lahmann, I., Brohl, D., Zyrianova, T., Isomura, A., Czajkowski, M.T., Kapoor, V., Griger, J., Ruffault, P.L., Mademtzoglou, D., Zammit, P.S., *et al.* (2019). Oscillations of MyoD and Hes1 proteins regulate the maintenance of activated muscle stem cells. *Gene Dev* 33, 524-535.
- Millay, D.P., Sutherland, L.B., Bassel-Duby, R., and Olson, E.N. (2014). Myomaker is essential for muscle regeneration. *Gene Dev* 28, 1641-1646.
- Paulsson, J. (2004). Summing up the noise in gene networks. *Nature* 427, 415-418.
- Raser, J.M., and O'Shea, E.K. (2005). Noise in gene expression: Origins, consequences, and control. *Science* 309, 2010-2013.
- Shimojo, H., Isomura, A., Ohtsuka, T., Kori, H., Miyachi, H., and Kageyama, R. (2016). Oscillatory control of Delta-like1 in cell interactions regulates dynamic gene expression and tissue morphogenesis. *Gene Dev* 30, 102-116.
- Sorensen, I., Adams, R.H., and Gossler, A. (2009). DLL1-mediated Notch activation regulates endothelial identity in mouse fetal arteries. *Blood* 113, 5680-5688.
- Vasyutina, E., Lenhard, D.C., Wende, H., Erdmann, B., Epstein, J.A., and Birchmeier, C. (2007). RBP-J (Rbpsi) is essential to maintain muscle progenitor cells and to generate satellite cells. *Proc Natl Acad Sci U S A* 104, 4443-4448.

Yoshioka-Kobayashi, K., Matsumiya, M., Niino, Y., Isomura, A., Kori, H., Miyawaki, A., and Kageyama, R. (2020). Coupling delay controls synchronized oscillation in the segmentation clock. *Nature*.

Reviewers' Comments:

Reviewer #1:

Remarks to the Author:

The authors have thoroughly addressed all of my concerns and the manuscript in my view is acceptable for publication.

Reviewer #2:

Remarks to the Author:

Authors provide compelling answers to all my remarks with important set of new experimental data. They also nicely extend their single cell model and entry/exit into quiescence or differentiation.

I am fully satisfied and recommend publication of the revised manuscript.

Reviewer #3:

Remarks to the Author:

The authors have addressed the raised comments properly, have re-written certain points in the manuscript for clarification and performed further experiments and analysis to support their conclusions. I agree with the authors that some suggested experiments might require establishment of new tools and procedures, which would take a substantial amount of time. These additional experiments would be nice-to-have, but are not absolutely essential to support the main points of the paper. Therefore, I fully support publication of their manuscript "Oscillations of Delta-like1 regulate the balance between differentiation and maintenance of muscle stem cells" in the revised form..

We thank all the reviewers for their time and effort to review our manuscript. We are pleased to see that the reviewers were overall satisfied with our revised manuscript.

REVIEWERS' COMMENTS

Reviewer #1 (Remarks to the Author):

The authors have thoroughly addressed all of my concerns and the manuscript in my view is acceptable for publication.

We thank this reviewer for the positive evaluation and for her/his comments on our original manuscript, which greatly strengthened our paper.

Reviewer #2 (Remarks to the Author):

Authors provide compelling answers to all my remarks with important set of new experimental data. They also nicely extend their single cell model and entry/exit into quiescence or differentiation.

I am fully satisfied and recommend publication of the revised manuscript.

We also thank this reviewer for the positive assessment and for her/his suggestion on our original manuscript. We addressed her/his points which strengthened our paper.

Reviewer #3 (Remarks to the Author):

The authors have addressed the raised comments properly, have re-written certain points in the manuscript for clarification and performed further experiments and analysis to support their conclusions. I agree with the authors that some suggested experiments might require establishment of new tools and procedures, which would take a substantial amount of time. These additional experiments would be nice-to-have, but are not absolutely essential to support the main points of the paper. Therefore, I fully support publication of their manuscript "Oscillations of Delta-like1 regulate the balance between differentiation and maintenance of muscle stem cells" in the revised form.

We thank the reviewer for the positive assessment and for her comments on our original manuscript, which we used to extend our manuscript and to strengthen its conclusion.